# The Truth Lies Somewhere in the Middle (of the Generated Tokens)

**Sophie L. Wang** [1]   **Phillip Isola** [1]   **Brian Cheung** [1]

## Abstract

How should hidden states generated autoregressively be collapsed into a representation that reflects a language model's internal state? Despite tokens being generated under causal masking, we find that mean pooling across their hidden states yields more semantic representations than any individual token alone. We quantify this through kernel alignment to reference spaces in language, vision, and protein domains. The improvement through mean pooling is consistent with information being distributed across generated tokens rather than localized to a single position. Furthermore, representations derived from generated tokens outperform those from prompt tokens, and alignment across generation reveals interpretable dynamics in model behavior.

**Project Page:** sophielwang.com/tokens
**Code:** github.com/sophicle/tokens

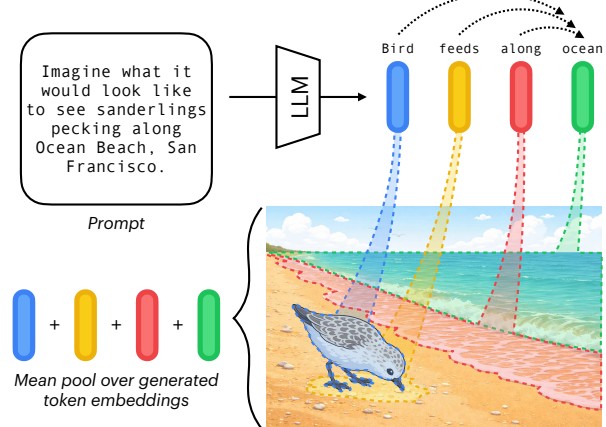

*Figure 1.* Conceptual illustration: Mean pooling across tokens generated by an autoregressive language model yields a representation that better captures the semantic content of the input than any individual token. The model is prompted to imagine the scene described by a caption. We compare the generated-token representations to reference representations of corresponding images.

## 1. Introduction

Representations in neural networks are used to interpret model performance and behavior, support downstream tasks such as information retrieval, and quantify relationships between what different intelligent systems learn (Azaria & Mitchell, 2023). In language models, these representations are often used as text embeddings that encode the semantic content of an input. The quality of these embeddings determines what information is available for downstream tasks and what aspects of the model's internal state can be studied (Reimers & Gurevych, 2019; Alain & Bengio, 2016). Thus, choosing how to extract an embedding is a methodological choice that affects both downstream task performance and model interpretability.

Prior work on language model representations considers hidden states extracted from a single forward pass over a

[1]MIT. Correspondence to: Sophie L. Wang <sophielw@mit.edu>.

*Proceedings of the $43^{rd}$ International Conference on Machine Learning*, Seoul, South Korea. PMLR 306, 2026. Copyright 2026 by the author(s).

fixed prompt. These hidden states span token, layer, and feature dimensions. To obtain a usable vector representation, these hidden states must be compressed. In bidirectional encoders, tokens are computed under shared context and are therefore comparable; hidden states are typically collapsed via final-token pooling or mean pooling (Neelakantan et al., 2022; Wang et al., 2023). In autoregressive models, however, averaging mixes hidden states computed under unequal context, and such representations perform poorly without modification (Jiang et al., 2024; BehnamGhader et al., 2024; Springer et al., 2025; Cheng et al., 2025; Fu et al., 2025; Lin et al., 2025; Zhang et al., 2025b; Hara et al., 2026).

It is unclear how the limitations of mean pooling for autoregressive models extend to generation, where hidden states are computed under a context increasingly determined by the model's own outputs (Vaswani et al., 2017). Generated tokens are not only outputs, but also inputs to later forward passes, so their hidden states may reflect semantic information accumulated over the generation. Because generative language models are not optimized to make their hidden states useful as embeddings, prior work trains models jointly for generation and representation (Muennighoff et al., 2025). This suggests that useful generative representations may re-

quire modifying the training objective. Yet, recent work also suggests that semantic information in autoregressive models can be distributed across the token trajectory (Liu et al., 2024) and that mean-pooled embeddings across generated tokens can faithfully capture instruction-following behavior (Wang et al., 2025). These observations leave open whether generated hidden states can yield semantic representations at inference time without additional modification, and if so, how they should be collapsed into a single embedding.

In this paper, we study representations derived from generated tokens by evaluating their alignment to reference spaces in language, vision, and protein domains. In Figure 1, we illustrate our finding that mean pooling across generated tokens yields more semantic representations than any individual token (Section 3.1). More generally, mixing token representations improves alignment because these tokens capture complementary information (Section 3.2). Additionally, representations derived from generated tokens are better aligned than those derived from prompt tokens, which do not exhibit the mixing phenomenon (Section 3.3). These results suggest that semantic information is distributed across generated tokens rather than localized to a single position (Section 3.4). We also demonstrate that representations across generation reveal a behavioral connection between the model's internal state and output tokens, such as recall (Section 3.5) and inconsistency (Section 3.6).

## 2. Methods

Last-token embedding

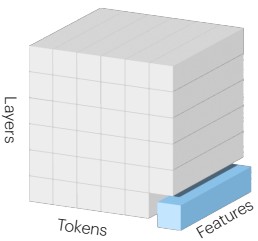

Token-mean embedding

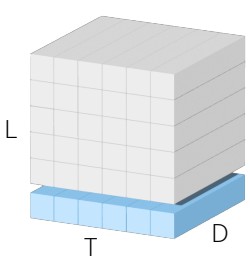

*Figure 2.* **Pooling the Token Dimension.** Each cube represents the activation tensor $H \in \mathbb{R}^{L \times T \times D}$ for a sample. Blue regions refer to the subset of activations that are selected and averaged across the token dimension to produce a $D$-dimensional embedding.

Our goal is to evaluate the semantic quality of representations derived from language model hidden states during generation. A useful representation for downstream tasks, such as retrieval or classification, preserves relationships among examples (Cristianini et al., 2001; Cortes et al., 2010). For instance, captions of visually similar images should be close together. We measure this by comparing the kernel induced by the language model representations to a reference kernel constructed from embeddings that encode relevant semantic structure (Kornblith et al., 2019; Sucholutsky et al., 2025).

For example, in vision-language experiments, the reference kernel is computed from image embeddings, so high alignment means that the language model's representations place image captions close together when their corresponding images are visually similar.

### 2.1. Preliminaries

Let $\{(x_i, y_i)\}_{i=1}^n$ be a fixed dataset of paired inputs. For each pair, we compute two representations: $u_i$ is the embedding produced from the language model input $x_i$, and $v_i$ is the embedding produced from the corresponding reference input $y_i$. We then compare how the two sets of embeddings organize the same $n$ samples. To do this, we form two similarity matrices, or kernels, $K, L \in \mathbb{R}^{n \times n}$, where

$$K_{ij} = u_i^\top u_j \quad \text{and} \quad L_{ij} = v_i^\top v_j.$$

Here, $K_{ij}$ measures how similar samples $x_i$ and $x_j$ are in the language model representation space, while $L_{ij}$ measures how similar the corresponding samples $y_i$ and $y_j$ are in the reference representation space. All embeddings are clipped at the 95th percentil of absolute feature values and then $\ell_2$-normalized before computing kernels.

Kernel alignment asks whether these two similarity structures agree. In other words, if two inputs are close according to the language model embeddings, then their paired reference inputs should also be close according to the reference embeddings. We quantify this using the debiased *Centered Kernel Alignment* (CKA) (Kornblith et al., 2019). Details are provided in Appendix A.

### 2.2. Language Model Representations

Given an input prompt $p$, an autoregressive language model generates a continuation of $T$ tokens; in the main text, $T = 128$. Each token is computed from the prompt and previous generated tokens through causal self-attention (Vaswani et al., 2017). We analyze hidden states at generated-token positions, excluding prompt-token states unless stated otherwise. Let $H \in \mathbb{R}^{L \times T \times D}$ denote the activation tensor, with $L$ layers and feature dimension $D$. Unless stated otherwise, we use final-layer states $h_t = h_{L,t} \in \mathbb{R}^D$ for $t = 1, \dots, T$.

**Pooling across tokens.** To obtain a fixed-dimensional vector representation from a sequence of tokens, we pool hidden states across token positions. We compare last-token pooling, which uses the final state of the trajectory, with mean-token pooling, which averages hidden states across the trajectory. These two choices provide a simple comparison between selecting one state and aggregating information over the full continuation, as visualized in Figure 2:

**Last-token pooling.** $\bar{h}_{\text{last}} = h_T$.

**Mean-token pooling.** $\bar{h}_{\text{mean}} = \frac{1}{T} \sum_{t=1}^{T} h_t$.

**Mixing token representations.** Mean pooling tests whether aggregating over all tokens of a particular sequence (e.g., prompt or generation) improves alignment, but it does not show whether the gain comes from a particular region or from combining multiple parts of the sequence. To study how semantic information is distributed across a token sequence, we analyze mixtures of representations drawn from distinct sources. For each sample, we construct three pooled representations $\bar{h}^{(1)}$, $\bar{h}^{(2)}$, $\bar{h}^{(3)}$, using the same pooling rule (last-token or mean pooling). The sources differ across experiments and may correspond to different token segments within a generation, different contiguous slices of tokens, or independent generations produced with different random seeds. We then form convex combinations

$$\bar{h}(\mathbf{w}) = \sum_{j=1}^{3} w_j \bar{h}^{(j)}, \qquad \mathbf{w} \in \Delta^2,$$

where $\Delta^2 = \{\mathbf{w} \in \mathbb{R}^3 : w_j \geq 0, \sum_j w_j = 1\}$ denotes the 2-simplex. Alignment is evaluated over a uniform barycentric grid on the simplex. In all experiments, we discretize $\Delta^2$ using $G = 20$ grid points per edge, yielding 210 weights.

**Prompts.** We choose prompts so that generation reflects the semantic structure we want to evaluate. The goal is not just to produce any continuation, but to put the model in the right task regime before extracting hidden states. In the vision-language setting, we follow Wang et al. (2025), who show that sensory prompts such as `Imagine what it would look like to see: {caption}.` make generative representations more aligned with vision models. We therefore use a visual cue, even though no image is given to the language model. We use the same idea in other domains. For reasoning tasks, prompts ask the model to solve the problem so that hidden states reflect the solution process rather than only the wording of the question. For protein-language tasks, prompts ask for biologically relevant descriptions so that representations reflect structural and functional properties. Prompt templates and sample generations are provided in Appendix B.1 and Appendix B.2.

### 2.3. Reference Model Representations

We evaluate language model representations by comparing them to fixed reference embeddings. For each paired sample $(x_i, y_i)$, the language model receives $x_i$, and the reference model embeds the corresponding target $y_i$. These reference embeddings define the semantic structure we want the language model representation to recover. For example, in vision–language experiments, two captions should be close in language space when their corresponding images are close in vision space.

Reference embeddings are held fixed across experiments, so changes in alignment reflect changes in the language model representation rather than changes in the target space. When reference embeddings have spatial or sequential axes, we mean-pool over the non-feature dimensions and use the final-layer representation. When the reference model is itself a language model, we use the same final-layer and token-pooling procedure described above.

The reference space differs by domain. In vision-language and protein-language tasks, the reference comes from an external object, such as an image or protein structure, whose structure is encoded by a pretrained model (Huh et al., 2024; Zhu et al., 2026; Edamadaka et al., 2025; Li & Walsh, 2026; Shu et al., 2025). In reasoning tasks, the reference comes from gold solutions, since there is no external modality. Alignment therefore measures whether generated-token representations recover the desired semantic structure: visual similarity for images, structural similarity for proteins, or solution similarity for reasoning. Following prior work on representational alignment, we use reference embeddings as a relational target that defines which samples should be close or far apart (Sucholutsky et al., 2025; Huh et al., 2024). Alignment measures whether the language model organizes the samples in the same way.

### 2.4. Models

We extract embeddings from a pretrained autoregressive language model and compare them against fixed reference embeddings. We use Qwen3-14B with thinking mode in the main text unless otherwise stated; **Qwen3** (Yang et al., 2025) is a decoder-only Transformer trained on large-scale multilingual and code data. Reference embeddings are obtained from pretrained encoders in other modalities. For vision, we use **DINOv2** (Oquab et al., 2024), a self-supervised Vision Transformer, and for protein structures, **ESM-3** (Hayes et al., 2025). Additional vision encoders are evaluated in Appendix Figure 23.

### 2.5. Datasets

We evaluate kernel alignment across vision-language, reasoning, and protein datasets. For vision-language, we use the **Wikipedia-based Image Text (WIT)** (Srinivasan et al., 2021) and the **Densely Captioned Images (DCI)** datasets (Urbanek et al., 2024), sampling 1024 examples from each. WIT is used in the main text unless otherwise stated. For reasoning-based evaluations, we use **Math-500** (Lightman et al., 2024) and the **GPQA Diamond** split (Rein et al., 2024). Prompts are constructed from problem statements, and reference embeddings are derived from gold solutions. For protein-language, we sample 1024 entries from the **UniProt** database (Consortium, 2024), using protein names in language model prompts and the corresponding protein structures for reference embeddings.

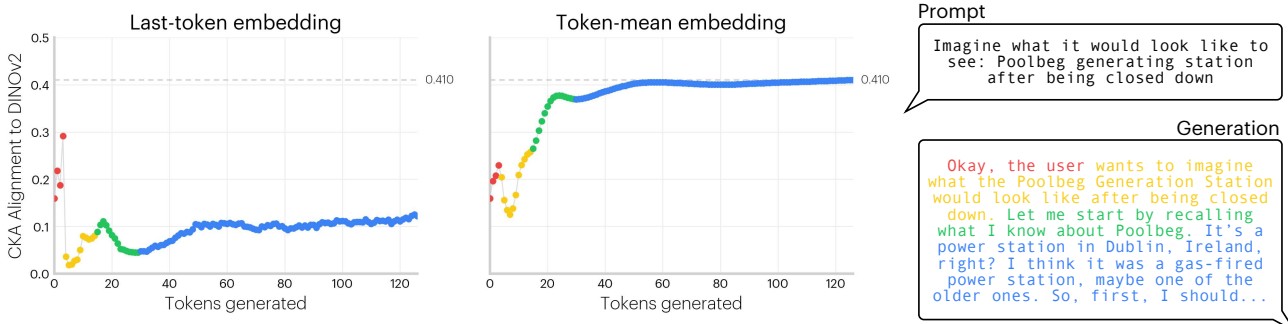

*Figure 3.* (*Left*) **Generated-token representations improve as tokens are averaged.** Vision–language alignment is quantified over 1024 samples using language model representations derived from last-token and mean-token embeddings during generation. The dashed line denotes the representation obtained by pooling across all tokens. Alignment increases as additional tokens are averaged and exceeds that of every individual token. Curves are averaged over five random seeds; variability across seeds is low (mean standard deviation $4.8 \times 10^{-3}$, max $9.6 \times 10^{-3}$), hence not visible in the figure. (*Right*) **Generated tokens induce representational phases.** Sample prompt and generation. Colored segments correspond to interpretable phases of the generation averaged over 1024 samples.

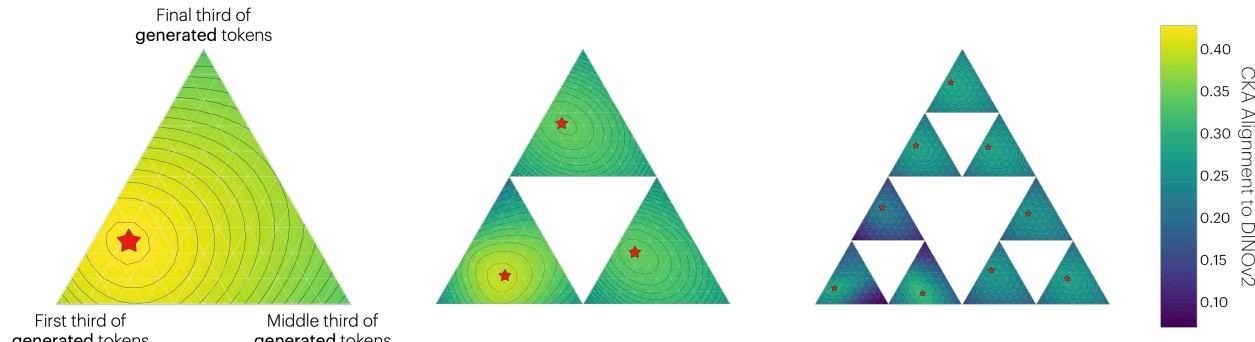

*Figure 4.* **Alignment is maximized by mixing generated tokens.** Vision–language alignment under convex combinations of token-slice ensembles at increasing levels of granularity. Across depths, alignment is consistently higher for interior convex combinations than for any single token slice, indicating that semantic information is distributed across the generation rather than localized to one segment. (*Left*) Depth 1: the simplex vertices correspond to embeddings computed by averaging hidden states over the first, middle, and final thirds of the generated token sequence. (*Middle*) Depth 2: each third is further subdivided into three contiguous token ranges, yielding simplices over finer-grained token slices. (*Right*) Depth 3: continued recursive subdivision of the token sequence into smaller contiguous ranges.

## 3. Results

### 3.1. Alignment Across Generated Tokens

We begin by studying kernel alignment across generated tokens. For each position, we compute token-level representations and evaluate their alignment to the reference space using the *last-token* and *token-mean* embeddings.

**Mean pooling generated tokens improves alignment.** In Figure 3, we find that alignment increases as additional tokens are incorporated, eventually exceeding that of any individual token. No single token attains maximal alignment. Alignment is also highly consistent across decoding seeds (mean standard deviation $4.8 \times 10^{-3}$, max $9.6 \times 10^{-3}$), indicating that the underlying semantic structure is stable despite variation in generated text. We show in Appendix Table 2 that mean pooling outperforms other pooling methods such as attention and max pooling.

**Alignment trajectories reveal phases of generation.** In Qwen3-14B, alignment changes systematically over the course of generation. We observe phases corresponding to (1) generic preamble, (2) prompt repetition, (3) recall, and (4) caption-specific response. Early tokens often contain task-generic text, while later tokens begin to retrieve information relevant to the prompt and then describe the specific scene. Correspondingly, alignment increases when the generation becomes more related to the underlying content. While the surface form varies across runs, this sequence produces similar alignment trajectories. The words produced during generation therefore give an interpretable trace of the model's computation, and kernel alignment measures how the corresponding hidden states move toward the reference structure. This connects to prior work showing that hidden representations can encode internal model states not directly available from the output (Burns et al., 2023; Azaria & Mitchell, 2023; Marks & Tegmark, 2023).

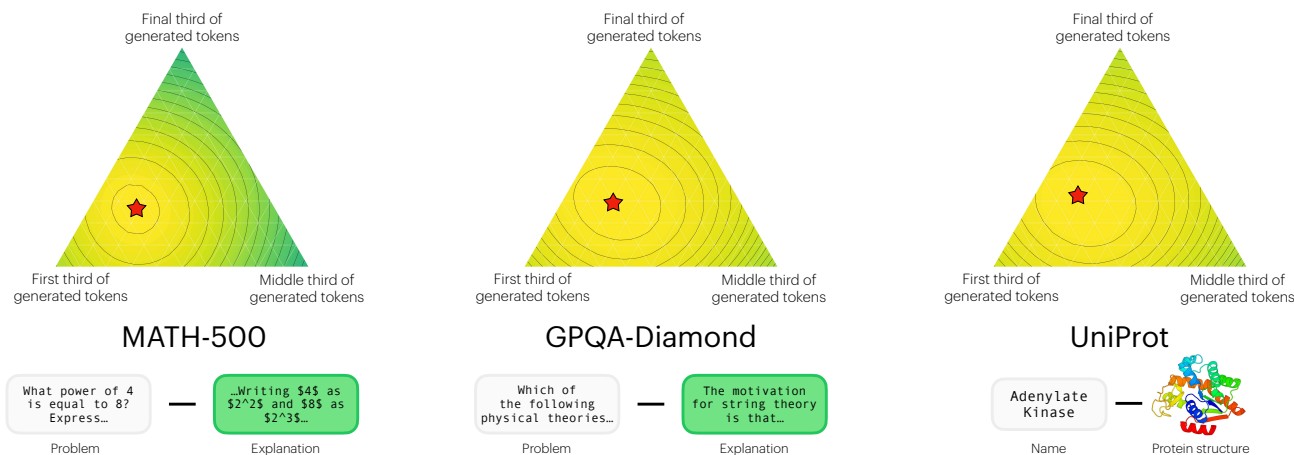

*Figure 5.* **Mixing generated-token representations improves alignment across reasoning and protein domains.** Each simplex shows kernel alignment under convex combinations of embeddings averaged over the first, middle, and final thirds of generated tokens. Across domains, alignment is maximized at interior combinations rather than at any single segment. This indicates that the mixing effect generalizes beyond vision–language alignment and appears in settings where the reference space reflects correctness or physical structure.

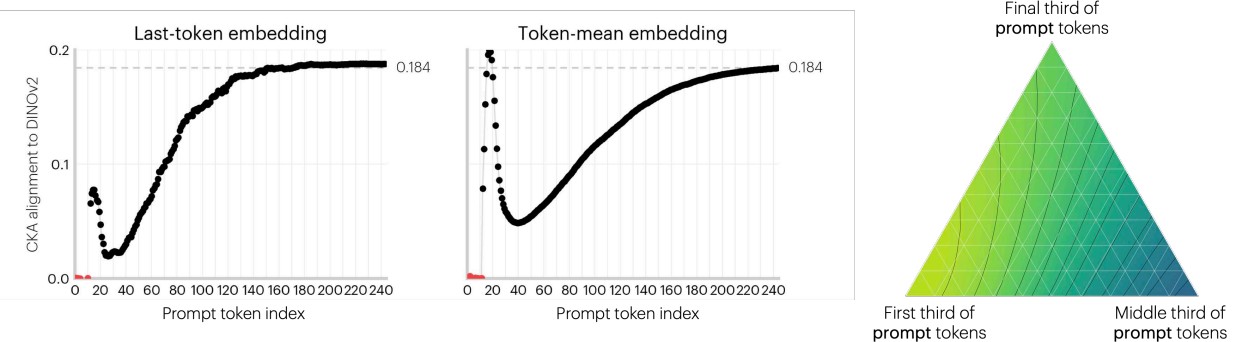

*Figure 6.* **Prompt-token representations do not benefit from mean pooling.** Extension of Figure 3 and Figure 4 to prompt tokens. Unlike generated-token representations, prompt-token representations do not improve under token averaging. The averaged prompt representation performs comparably to the best individual prompt tokens near the end of the prompt, but does not exceed them. Red denotes the tokens corresponding to `Imagine what it would look like to see.`

## 3.2. Mixing Generated Tokens

We next ask whether the gains from mean pooling reflect a more general property. While averaging across all tokens outperforms any individual token, it is unclear whether mixing representations from arbitrary subsets of tokens also improves alignment. We evaluate vision–language alignment under convex combinations of embeddings derived from contiguous segments of a generation. For each sample, we construct base representations by mean pooling over token segments (e.g., first, middle, and final thirds), and recursively subdivide these segments to obtain finer partitions. Alignment is then evaluated over the simplex of convex combinations.

**Mixing generated tokens improves alignment.** As shown in Figure 4, alignment is consistently higher in the interior of the simplex, where representations are mixed, than at any individual segment. Thus, the benefit is not limited to averaging over the full generation: more generally, combining representations from different token regions improves alignment. This suggests that different parts of the generation carry complementary semantic information, rather than a single segment containing the full semantic structure. The pattern persists across levels of granularity, indicating that alignment is not localized to one region of the generation. The geometry of these mixtures also varies with token position. At finer partitions, earlier segments exhibit less uniform mixing than later segments, likely because early generated tokens already have relatively high individual alignment (Figure 13a). Later segments show a flatter mixing geometry, suggesting that their contribution is more evenly distributed across neighboring token regions.

**Mixing generated tokens yields improvement in alignment in reasoning and protein domains.** We observe the same structure in reasoning and protein domains. On

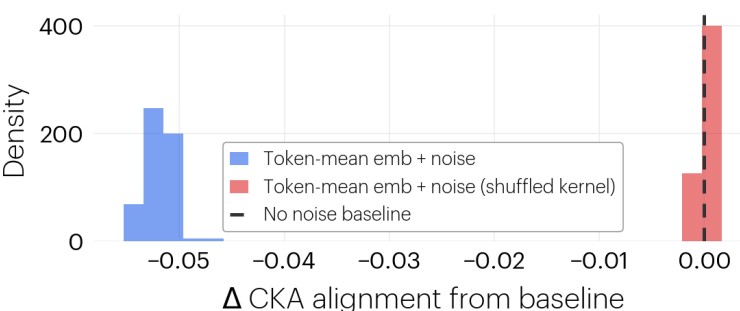

*Figure 7.* **Averaging is a confounder for alignment between unrelated kernels.** We sample 100 isotropic noise perturbations ($\epsilon = 1.0$) around the averaged token representation and plot the resulting distribution of alignment (gray). The dashed vertical line indicates the averaged representation without noise. (*Left*) Original image-text pairings. (*Right*) Image-text pairings are shuffled, breaking semantic correspondence.

MATH-500 and GPQA-Diamond, the reference space is derived from gold solutions, so improved alignment suggests that mixing better captures correctness-relevant structure. On UniProt, the reference space is derived from protein structure, so the same effect indicates better agreement with a physically grounded target. Across these settings, mixing token representations again yields higher alignment than any individual segment (Figure 5). This suggests that distributed semantic information across generated tokens is not specific to visual descriptions, but appears across tasks with different forms of reference structure.

### 3.3. Alignment Across Prompt Tokens

We compare generated-token representations to prompt-token representations because the prompt already contains the semantic content being evaluated. If generation only supplies additional positions to average over, prompt-token representations should recover similar structure. If generated tokens are better aligned, this would suggest that generation changes how the input is represented, rather than merely providing more hidden states.

**Generated tokens produce better representations than prompt tokens.** Generated-token representations have higher alignment than prompt-token representations, both for individual tokens and after mean pooling (0.410 vs. 0.184). This is surprising because both representations are derived from the same caption. The prompt already contains the relevant information, but extracting hidden states from the prompt does not make this information as accessible in representation space. Generation appears to make the same semantic content easier to recover. As the model produces a continuation, its hidden states become better aligned with the reference structure tied to the original caption. Thus, the generated text helps the model form a representation that better reflects the prompt's content. Consistent with this, mean-pooled generated representations outperform prompt-based representations on retrieval, ranking, and clustering (Tables 3–5).

**Mean pooling prompt tokens does not improve alignment.** In Figure 6, we study representations derived from

prompt tokens. Unlike generated-token representations, prompt-token representations do not improve under token averaging. Later prompt tokens are often comparable to the mean-pooled prompt-token representation. This is consistent with the causal masking limitation of decoder-only embeddings: earlier prompt tokens cannot access later tokens, so averaging over prompt positions can dilute information rather than aggregate it (Springer et al., 2025).

### 3.4. Why Does Mixing Improve Alignment?

**Improvements are not explained by averaging alone.** A potential confound is that averaging may increase alignment because of variance reduction. To test this, we treat individual token embeddings as noisy samples of an underlying direction. We do this by sampling isotropic noise perturbations around the token-mean representation and comparing their alignment to the unperturbed mean (Figure 7). When image-text correspondence is preserved, the mean achieves higher alignment as we find in Section 3.1. When correspondence is broken by shuffling image-text pairings, this improvement disappears. These results show that averaging improves alignment only when the pooled representation is already centered on meaningful semantic structure. Variance reduction on noisy samples of unrelated kernels alone therefore cannot explain why pooling across generated tokens yields better representations (Section 3.7).

**Mixing captures complementary information.** Alignment improves when combining representations from multiple tokens, rather than using any single token alone. This suggests that individual tokens do not simply accumulate all semantic information over generation, even though later tokens are conditioned on earlier ones. Instead, different tokens capture complementary aspects of the same underlying content, so combining them yields stronger representations.

Evidence for this interpretation comes from both prior work and our experiments. Springer et al. (2025) show that repeating prompts improves embeddings for fixed inputs, suggesting that representations derived under related contexts can carry complementary information. In our setting, we observe a similar effect across independent generations of

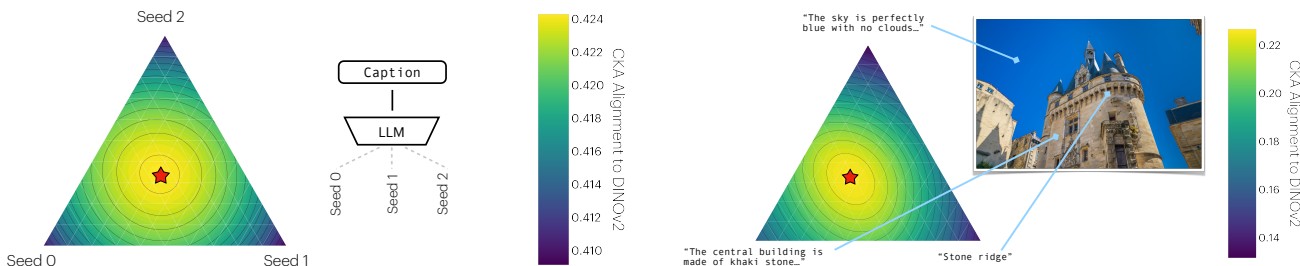

*Figure 8.* (*Left*) **Mixing representations across decoding seeds improves alignment.** Convex combinations of generation embeddings from three independent decoding runs of the same input. Vertices correspond to different random seeds. (*Right*) **Mixing representations across different views of a scene improves alignment.** Convex combinations of region-conditioned generation embeddings. Vertices correspond to three region-level captions for the same image. In both cases, alignment is maximized at interior convex combinations rather than at any single vertex, indicating that pooling integrates complementary information across independent generations or semantic views.

the same prompt. As shown in Figure 8 (left), convex combinations of embeddings from different decoding seeds consistently achieve higher alignment than any single run. The effect is strongest when representations correspond to explicitly different views: on the DCI dataset (Figure 8, right), combining region-conditioned generations that describe distinct parts of an image yields higher alignment than any individual region-conditioned representation.

These findings connect to prior work on linear compositionality in embedding spaces, where combining representations can produce meaningful semantic structure (Mikolov et al., 2013; Pennington et al., 2014; Arora et al., 2017; Elhage et al., 2022; Tigges et al., 2024). Our results suggest that this compositionality extends to token-level representations within a single autoregressive generation. This is not obvious a priori, since generated-token embeddings are typically treated as transient states.

Recent work on mean pooling argues that prompt-side text embeddings can remain informative when different texts have sufficiently distinct mean embeddings and token embeddings within each text are concentrated around their mean (Hara et al., 2026). This explains why mean pooling need not degrade prompt-token representations: when means are already distinct, averaging preserves the relevant first-order structure. However, this account does not explain why alignment *improves* as additional generated tokens are averaged. In our setting, averaging does not merely preserve an informative prompt representation; it moves the accumulated representation toward a region of feature space that better captures the underlying semantics of the prompt.

### 3.5. Representational Phases During Generation

As shown in Figure 3, kernel alignment across tokens, both through last-token and mean-token pooling, exhibits reproducible phase structure. We examine these phases more directly by intervening on the generated tokens and asking whether specific phrases can induce the same representa-

tional shifts even when their semantic content is minimal.

**Certain phrases elicit interpretable changes in representation space.** In generations with explicit thinking traces, the model produces generic phrases such as "Okay, the user wants..." or "Let me recall what I know." These output tokens alone do not carry information that is semantically relevant to the prompt. However, we find that such phrases are consistently associated with sharp changes in alignment, suggesting that they correspond to transitions in the model's internal state. This complements the finding that generic tokens such as "Hmm" or "Wait" generated during reasoning can exhibit a significant increase in mutual information to the correct answer (Qian et al., 2026).

We test whether this effect can be induced by injecting tokens (Figure 9). We disable thinking mode so that the model produces a shorter, more direct answer (e.g., an output that begins with "Visualizing the Poolbeg Generating Station after being closed down..."). We then prepend the phrase "Let me recall what I know" to the beginning of this generation. Despite the phrase carrying no scene-relevant information, this intervention induces a clear spike in alignment.

Although "let me recall" is generic, it corresponds to a shift in the model's internal representation toward the reference representation, occurring before any scene-relevant information appears in the output. This indicates that the recall phase induces a corresponding shift in representation space. More broadly, this suggests that kernel alignment can serve as a causal probe of internal state.

### 3.6. Model-Specific Representation Dynamics

We study how mismatching the generating model and the embedding model affects representations. To do so, we separate generation from embedding: we generate text using a stronger model (Qwen3-14B) and a weaker model (OLMo3-7B-Think), and compute mean-pooled representations in both models' embedding spaces.

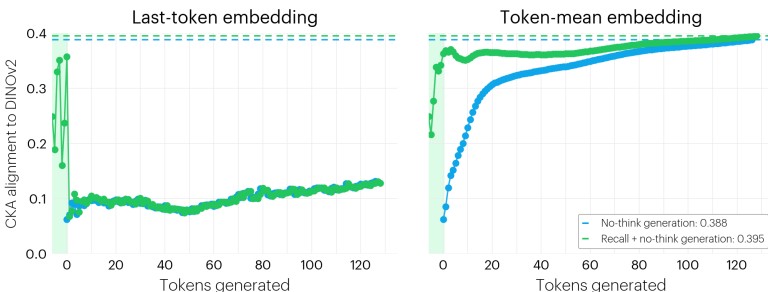

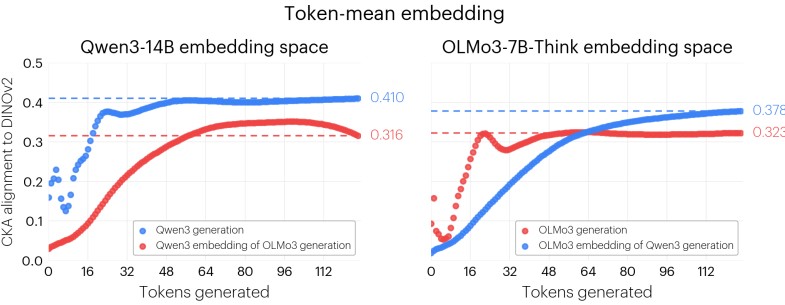

*Figure 9.* **A generic recall phrase induces a spike in alignment.** Tokenwise alignment when a recall phrase is injected into generation without thinking mode. The dashed line denotes the representation obtained by pooling across all tokens. Prepending the phrase "Let me recall what I know" (highlighted green region) induces a sharp alignment spike. This suggests that recall-like tokens can shift the model's representational state even before scene-relevant content appears in the output.

*Figure 10.* **Representational dynamics depend on the embedding model.** Token-wise alignment computed across generating and embedding models. The dashed line denotes the representation obtained by pooling across all tokens. Responses are generated by Qwen3-14B and OLMo3-7B-Think. Red: OLMo3 generations; blue: Qwen3 generations. The disappearance of the original phase structure under mismatched generation and embedding suggests that these dynamics are not properties of the text alone, but depend on how a model processes its own generated tokens.

**Phase structure disappears when generations are embedded by a different model.** We find in Figure 10 that when a model embeds text that it did not generate, the representational phase structure observed in Figure 3 disappears. This indicates that the observed phase structure is not a property of the text alone. Instead, it depends on how a particular model's internal representation evolves as it generates. Phrases that correspond to clear transitions in one model's representation space do not induce the same transitions when processed by another model. Thus, the phase structure observed when a model embeds its own generations appears functionally tied to its internal computation.

**Tokens inconsistent with model knowledge degrade its representation.** When alignment is measured in Qwen3's embedding space (Figure 10, left), representations from Qwen3's own generations improve monotonically as more tokens are averaged. In contrast, Qwen3 embeddings of OLMo3-generated text initially improve but then degrade. We hypothesize that later tokens introduce information inconsistent with Qwen3's internal state. For example, when prompted with the caption `Poolbeg generating station after being closed down`, OLMo3 continues: "First, I need to recall what Poolbeg Generating Station is. I think it's a power plant in Canada, maybe in Ontario? Let me confirm that. Yeah, I believe Poolbeg is near Toronto, specifically in the province of Ontario." This is incorrect: Poolbeg Generating Station is in Dublin,

Ireland, as Qwen3 notes in its own generation. While such continuations may be internally consistent for OLMo3, they diverge from the visual content and no longer improve alignment in Qwen3's embedding space. Conversely, in OLMo3's embedding space (Figure 10, right), OLMo3's generations benefit from averaging, suggesting that OLMo3 treats its own output as internally consistent. Notably, OLMo3 embeddings of Qwen3-generated text yield even higher alignment than embeddings of OLMo3's own generations, suggesting that the stronger model's text better reflects the underlying visual state.

### 3.7. Alignment Across Layer Depth

**Mean pooling across layer depth does not improve alignment.** So far, we have used final-layer hidden states to construct generated-token representations. We now ask whether the benefit of averaging is specific to tokens, or whether a similar effect appears when averaging across network depth. For each transformer layer, we first mean-pool hidden states across the token dimension to obtain a layer-specific representation. We then compare these representations to those obtained by additionally averaging across layers. As shown in Figure 11, averaging across layers yields alignment comparable to the best single layer, but does not improve upon it. This is consistent with Nguyen et al. (2020), who find redundancy among representations across depth. We also find that late-layer representations achieve high alignment,

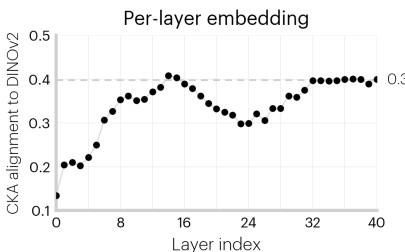 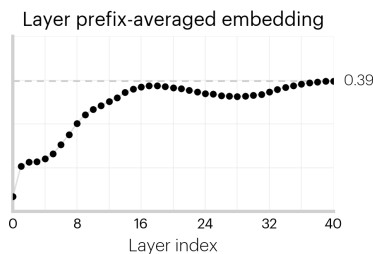

*Figure 11.* **Averaging across layers does not improve representations.** Vision-language alignment as a function of network depth for Qwen3-14B generated-token representations aligned to DINOv2. The dashed line denotes the representation obtained by pooling across all layers. Layer averaging performs comparably to the best single layer, but does not improve beyond it. In contrast, Appendix Figure 25 shows that prompt-token representations are best at intermediate layers.

in contrast to prior work on prompt-side embeddings where intermediate layers often perform best (Skean et al., 2025; Barbero et al., 2025; Jin et al., 2025; Gurnee & Tegmark, 2024; Fan et al., 2025). Appendix Figure 25 shows that this intermediate-layer advantage holds for prompt-token representations but not for generated tokens, suggesting that representations formed during generation behave differently. We leave this discrepancy as an open question.

## 4. Implications

We find that hidden states formed during autoregressive generation are best collapsed into a vector representation by mean pooling across tokens, rather than selecting any single token. We discuss further implications of our work.

**Generation yields better text embeddings.** Section 3.3 shows that continuations produced by an autoregressive model provide hidden states that can be mean-pooled to produce better text embeddings than those derived from prompt embeddings. Existing work on decoder-only embeddings often treats causal masking as a limitation of mean pooling, motivating additional methods such as prompt repetition or inference-time modifications (Springer et al., 2025; Jiang et al., 2024; Cheng et al., 2025; Fu et al., 2025). We show that generation provides another source of context. The model's own continuation makes the input's semantic content more accessible in hidden-state space. Averaging hidden states across generated tokens therefore offers a simple way to extract embeddings that better reflect the caption.

**Generated tokens induce representational state changes.** Generated tokens are not only outputs of the model; in an autoregressive model, they also become part of the context used to compute later hidden states (Pal et al., 2023). This makes generation a setting in which outputs and internal states are coupled over time. Section 3.5 shows that particular generated phrases, such as recall-related tokens, correspond to shifts in alignment before scene-specific information is explicitly produced. This suggests that token-level outputs can help induce changes in the model's representational state, and that alignment to reference spaces provides

a way to study these state changes (Zhang et al., 2025a; Afzal et al., 2025). This connects to recent work on reasoning dynamics, where "thinking tokens" such as "Hmm" or "Wait" correspond to sharp increases in mutual information with the correct answer (Qian et al., 2026).

**Hidden states across generation behave differently.** Finally, generated hidden states should not be treated as a straightforward extension of prompt-side representations. Prompt hidden states are computed under a fixed external context, while generated hidden states are computed under a context increasingly produced by the model itself. This difference matters empirically because we find that prompt-token averaging does not show the same convex representation structure. This supports a trajectory-based view of autoregressive representations, where semantic information is distributed across continuations rather than localized to a single hidden state (Liu et al., 2024).

## 5. Limitations

Generative representations require autoregressive decoding, introducing a computational tradeoff relative to standard embeddings. Prompt embeddings require a single forward pass, while generative representations require producing $T$ tokens, with cost scaling linearly in $T$. Our analysis uses kernel alignment to reference representations, which provides an aggregate measure of semantic similarity across samples. Since this operates on relational structure rather than individual embeddings, alignment reflects average consistency with a reference space rather than correctness for any specific generation. Finally, averaging hidden states across tokens does not produce an embedding that can be fed back into the model to continue generation as "soft" concept tokens can be (Hao et al., 2024; Zhang et al., 2026), likely because such averages fall outside the distribution of states encountered during decoding. These representations should therefore be interpreted as probes of internal state rather than usable generative states.

## Impact Statement

This paper presents work whose goal is to advance the field of Machine Learning. There are many potential societal consequences of our work, none of which we feel must be specifically highlighted here.

## Acknowledgements

We are grateful to Antonio Norelli for the idea of mixing representations across decoding seeds (Figure 8). We also thank Amil Dravid and Kento Nishi for writing feedback. This work was supported by the DARPA Mathematics for the DIscovery of ALgorithms and Architectures (DIAL) program, the DARPA Knowledge Management at Scale and Speed (KMASS) program, the NSF award 2124052, the Air Force Office of Scientific Research (AFOSR) under award number FA9550-21-1-0399, a Packard Fellowship to P.I., and by ONR MURI grant N00014-22-1-2740.

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

## A. Kernel alignment metrics

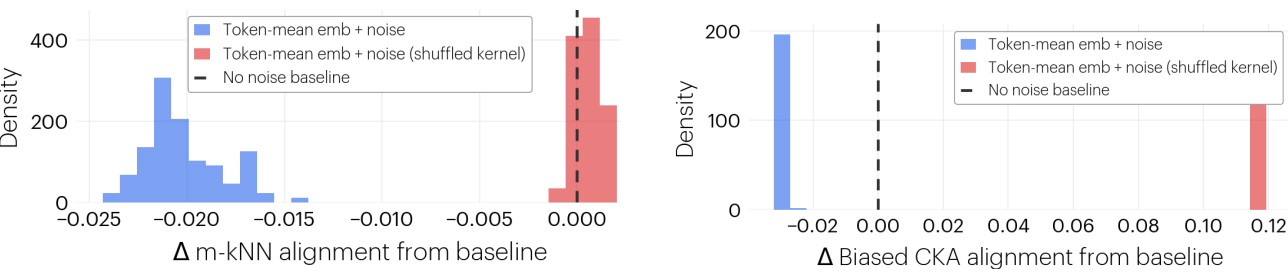

*Figure 12.* Extension of Figure 7 to mutual-$k$NN alignment metric with $k = 10$ and biased CKA alignment metric.

We define our choice of alignment metrics and validate that the qualitative phenomena reported in the main text are not artifacts of a particular similarity measure. In particular, we justify the use of *debiased* centered kernel alignment (CKA) as our primary metric, and show that our main trends persist under a neighborhood-based alignment measure, mutual $k$-nearest neighbors (m-$k$NN).

**Centered Kernel Alignment (CKA).** Let $\{u_i\}_{i=1}^n \subset \mathbb{R}^{d_u}$ and $\{v_i\}_{i=1}^n \subset \mathbb{R}^{d_v}$ denote two sets of representations. We define the uncentered Gram matrices $K, L \in \mathbb{R}^{n \times n}$ by

$$K_{ij} = \langle u_i, u_j \rangle, \qquad L_{ij} = \langle v_i, v_j \rangle.$$

The standard, biased version of linear CKA uses ordinary centered Gram matrices (Kornblith et al., 2019). Let

$$H = I_n - \frac{1}{n}\mathbf{1}\mathbf{1}^\top, \qquad K_c = HKH, \qquad L_c = HLH.$$

The biased HSIC estimator is (Gretton et al., 2005)

$$\text{HSIC}_{\text{biased}}(K, L) = \frac{1}{(n-1)^2} \operatorname{tr}(K_c L_c),$$

and biased CKA is

$$\text{CKA}_{\text{biased}}(K, L) = \frac{\text{HSIC}_{\text{biased}}(K, L)}{\sqrt{\text{HSIC}_{\text{biased}}(K, K)\text{HSIC}_{\text{biased}}(L, L)}}.$$

However, biased CKA can produce inflated similarity values when the number of features is large relative to the number of samples, even for unrelated or random representations (Davari et al., 2023; Murphy et al., 2024). For this reason, we use debiased CKA, which replaces ordinary centering with the unbiased U-centering operation (Song et al., 2012; Murphy et al., 2024). First set the diagonal entries of $K$ and $L$ to zero. For a hollow Gram matrix $A \in \mathbb{R}^{n \times n}$ with $n > 2$, define its U-centered version $A^U$ by

$$A_{ij}^U = \begin{cases} A_{ij} - \frac{1}{n-2}\sum_{\ell=1}^n A_{i\ell} - \frac{1}{n-2}\sum_{k=1}^n A_{kj} + \frac{1}{(n-1)(n-2)}\sum_{k,\ell=1}^n A_{k\ell}, & i \neq j, \\ 0, & i = j. \end{cases}$$

Let $K^U$ and $L^U$ denote the U-centered versions of $K$ and $L$. The unbiased HSIC estimator can then be written as

$$\text{HSIC}_{\text{unbiased}}(K, L) = \frac{1}{n(n-3)} \sum_{i \neq j} K_{ij}^U L_{ij}^U, \qquad n > 3.$$

Equivalently, since $K^U$ and $L^U$ are hollow,

$$\text{HSIC}_{\text{unbiased}}(K, L) = \frac{1}{n(n-3)} \operatorname{tr}(K^U L^U).$$

We define debiased CKA as the normalized unbiased HSIC:

$$\text{CKA}_{\text{debiased}}(K, L) = \frac{\langle K^U, L^U \rangle_F}{\sqrt{\langle K^U, K^U \rangle_F \langle L^U, L^U \rangle_F}}.$$

Unlike biased CKA, debiased CKA can take values in $[-1, 1]$.

Throughout the main text, we report alignment using debiased CKA. As illustrated in Figure 12, adding isotropic noise to the token-mean representation or breaking semantic correspondence via kernel shuffling leads to increased alignment under biased CKA, despite the absence of meaningful correspondence. In contrast, debiased CKA remains centered around the no-noise baseline (Figure 7), correctly reflecting the loss of semantic structure. This distinction is critical for interpreting the convex alignment effects studied in Section 3.2.

**Mutual $k$-Nearest Neighbors (m-$k$NN) alignment.** Given representations $\{u_i\}_{i=1}^n$ and reference representations $\{v_i\}_{i=1}^n$, we define for each sample $i$ the $k$-nearest-neighbor sets $\mathcal{N}_k^u(i)$ and $\mathcal{N}_k^v(i)$ under cosine distance. The m-$k$NN alignment score is

$$\text{m-kNN} = \frac{1}{n} \sum_{i=1}^n \frac{|\mathcal{N}_k^u(i) \cap \mathcal{N}_k^v(i)|}{k}.$$

Mutual $k$NN was previously used to study representational convergence across modalities with increasing model scale (Huh et al., 2024). By emphasizing overlap in local neighborhood structure rather than global similarity, m-$k$NN provides a more permissive notion of alignment.

As shown in Figure 13, we recover the same qualitative patterns under m-$k$NN as in the main text: token-mean representations outperform individual tokens, alignment improves under convex combinations of token slices, and interior simplex points achieve higher alignment than vertices. This confirms that our conclusions do not depend on a specific alignment metric.

Despite this robustness, we do not adopt m-$k$NN as our primary metric. In practice, m-$k$NN is sensitive to sampling noise, particularly in low-sample regimes such as GPQA-Diamond and MATH-500, and depends on the choice of $k$, which acts as a heuristic hyperparameter. For these reasons, we report debiased CKA in the main text and use m-$k$NN only as a corroborating analysis.

## B. Experiment details

We document the prompting formats and representative generations used throughout the paper to specify the exact inputs used to elicit generative representations across task families, and to provide verbatim examples of model outputs, illustrating the kinds of text over which token-level and pooled representations are computed.

### B.1. Prompting templates

We use task-specific prompt templates for all generative experiments. Prompts are designed to elicit grounded descriptions, factual recall, or structured reasoning, depending on the task domain.

For image-caption datasets (WIT and DCI), we adopt the prompt formulation introduced by Wang et al. (2025):

```
Imagine what it would look like to see: {caption}.
```

For protein names on the UniProt dataset, we use:

```
Provide a thorough summary of {protein name}.  Include its gene name,
protein family, molecular weight, known structural domains, function in the
cell, binding sites, any known interactions or pathways it participates in.
```

For open-ended question answering tasks, including MATH-500, we use:

```
Solve the following problem and give the correct answer:  {problem}
```

For multiple-choice question answering tasks, such as GPQA-Diamond, we use:

```
Solve the following problem and output answer as \boxed{A/B/C/D}.  {problem}
```

For multiple-choice tasks, the final answer is extracted by parsing the model output and selecting the option appearing inside the final \boxed{} expression.

### B.2. Sample generations

We present representative 128-token generations from Qwen3-14B for each task family, using the prompt templates described above. Generations are shown verbatim, without post-processing, truncation (other than the token limit), or manual correction, except where explicitly noted.

These examples are intended to illustrate the qualitative structure of model outputs, including reasoning traces, recall phases, and domain-specific content, rather than to serve as evidence for any particular quantitative result.

---

**Image–caption generation (WIT / DCI)**

**Prompt:**
```
Imagine what it would look like to see:  Poolbeg generating station after
being closed down
```

**Model:** Qwen3-14B

**Generation:**
```
<think>
Okay, the user wants to imagine what the Poolbeg Generating Station would
look like after being closed down.  Let me start by recalling what I know
about Poolbeg.  It's a power station in Dublin, Ireland, right?  I think
it was a gas-fired power station, maybe one of the older ones.  So, first,
I should confirm some details about its history and current status.  Wait,
I believe it was decommissioned a few years ago.  Let me check that.  Oh
right, it was closed in 2017, I think.  So the user is asking about its
appearance post-closure.
Now
```

---

---

Protein annotation (UniProt)

**Prompt:**
Provide a thorough summary of HUMAN SIGMA ALCOHOL DEHYDROGENASE. Include
its gene name, protein family, molecular weight, known structural domains,
function in the cell, binding sites, any known interactions or pathways it
participates in.

**Model:** Qwen3-14B

**Generation (excerpt):**
```
<think>
Okay, so I need to provide a thorough summary of Human Sigma Alcohol
Dehydrogenase. Let me start by recalling what I know about alcohol
dehydrogenases. I remember that there are different classes of alcohol
dehydrogenases, like the classical ADHs which are found in the liver. But I
think Sigma is a different isoform.
First, the gene name. I think the main ADH genes are ADH1A, ADH1B, ADH1C,
etc. Maybe the sigma subunit is part of a different family. Wait, the
sigma isoform might be part of the AD
```

---

Open-ended question answering (MATH-500)

**Prompt:**
Solve the following problem and give the correct answer: How many positive
whole-number divisors does 196 have?

**Model:** Qwen3-14B

**Generation (excerpt):**
```
<think>
Okay, so I need to figure out how many positive whole-number divisors 196
has. Hmm, let me think. I remember that to find the number of divisors
of a number, you first have to find its prime factorization. Then, you add
one to each of the exponents in the prime factorization and multiply those
together. Is that right? Let me make sure.
For example, if a number is factored into primes like 2^a * 3^b * 5^c...,
then the number of divisors is (a+1)*(b+1)*(c
```

---

Multiple-choice question answering (GPQA-Diamond)

**Prompt:**
```
Solve the following problem and output answer as \boxed{A/B/C/D}.  ChIP-seq
on a PFA-fixed sample with an antibody to the IKAROS transcription factor
in human B cells followed by next-generation sequencing and standard
quality control, alignment and peak-calling steps produced ChIP peaks that
disappeared when PFA+DSG fixation was used.  Where are we most likely to
find such disappearing peaks?
Choices:  A. In the introns of large genes B. At active promoters and
enhancers C. At random locations in the genome D. At repeats
```

**Model:** Qwen3-14B

**Generation:**
```
<think>
Okay, so I need to figure out where the disappearing ChIP peaks are most
likely found when comparing PFA fixation alone versus PFA+DSG fixation.
Let me start by recalling what I know about ChIP-seq and the role of
crosslinkers like PFA and DSG.
ChIP-seq is used to study protein-DNA interactions, like transcription
factors binding to specific genomic
```

For multiple-choice tasks, the predicted answer is extracted by parsing the option appearing inside the final `\boxed{}` expression.

### B.3. Detailed experiment settings

Generations are produced using sampling decoding with automatic dtype selection (`torch_dtype=auto`). Unless otherwise specified, we rely on each model's default generation configuration for temperature, top-$k$, and top-$p$. All models are used in evaluation mode without finetuning.

*Table 1.* LLM checkpoints used in experiments.

| Model | Checkpoint |
|---|---|
| Qwen3 8B | Qwen3-8B |
| Qwen3 14B | Qwen3-14B |
| Olmo3 7B | Olmo3-7B-Think |
| gpt-oss 20B | gpt-oss-20b |

## C. Tokenwise

We present additional analyses and ablations that support and clarify the main findings. We use these experiments to test alternative explanations for the observed token-wise and pooled alignment effects, including whether they depend on the choice of alignment metric, token order, sample correspondence, model architecture, or representational scale. Each figure addresses one of these possibilities.

**Metric robustness.** Figure 13 extends the main tokenwise and convex-combination results to the m-$k$NN alignment metric. The top panel (Figure 13a) shows that the qualitative structure of tokenwise alignment trajectories, including monotonic improvement under prefix averaging and reproducible phase structure, is preserved under m-$k$NN. The bottom panel (Figure 13b) shows that alignment under convex combinations of token-slice representations continues to be maximized at interior simplex points across increasing levels of granularity. These results confirm that the observed convex structure is not an artifact of CKA and holds under a neighborhood-based similarity measure.

**Role of token order and sample correspondence.** Figure 14 presents three shuffling ablations that isolate distinct sources of structure in the alignment curves. In the top row, token embeddings are shuffled *after* embedding, preserving

| Method | CKA ↑ | m-kNN ↑ |
|---|---|---|
| Mean pool | 0.4106 | 0.1543 |
| Best per-token | 0.2910 | 0.0914 |
| Attention pool | 0.2932 | 0.0981 |
| Max pool | 0.3505 | 0.1239 |

*Table 2.* Extension of Figure 3: Alignment across pooling methods.

| Method | R@1 ↑ | R@5 ↑ | R@10 ↑ |
|---|---|---|---|
| Generation mean pool | **0.052** | **0.100** | **0.121** |
| Caption mean pool | 0.027 | 0.065 | 0.078 |
| Caption last token | 0.022 | 0.048 | 0.062 |
| Generation last token | 0.008 | 0.020 | 0.029 |

*Table 3.* Structure-based retrieval benchmark for WiT seed 0. R@k measures the average overlap between the top-$k$ language neighbors induced by each pooling rule and the top-$k$ DINOv2 vision neighbors for the same example.

| Method | MRR ↑ | Mean Rank ↓ | Median Rank ↓ |
|---|---|---|---|
| Generation mean pool | **0.342** | **21.0** | **7** |
| Caption mean pool | 0.263 | 34.4 | 12 |
| Caption last token | 0.210 | 43.8 | 19 |
| Generation last token | 0.113 | 65.3 | 39 |

*Table 4.* For each query, the relevant set is the top-10 vision neighbors; the table reports the rank of the first relevant hit in the language-side ranking, summarized as MRR, mean rank, and median rank.

| Method | ARI mean ↑ | NMI mean ↑ |
|---|---|---|
| Generation mean pool | **0.095** | **0.311** |
| Caption mean pool | 0.040 | 0.213 |
| Caption last token | 0.045 | 0.222 |
| Generation last token | 0.006 | 0.130 |

*Table 5.* Separate KMeans models are fit in language and vision spaces at $k \in \{10, 20, 50\}$ after row normalization, and the table reports the mean adjusted Rand index and normalized mutual information across those three cluster counts.

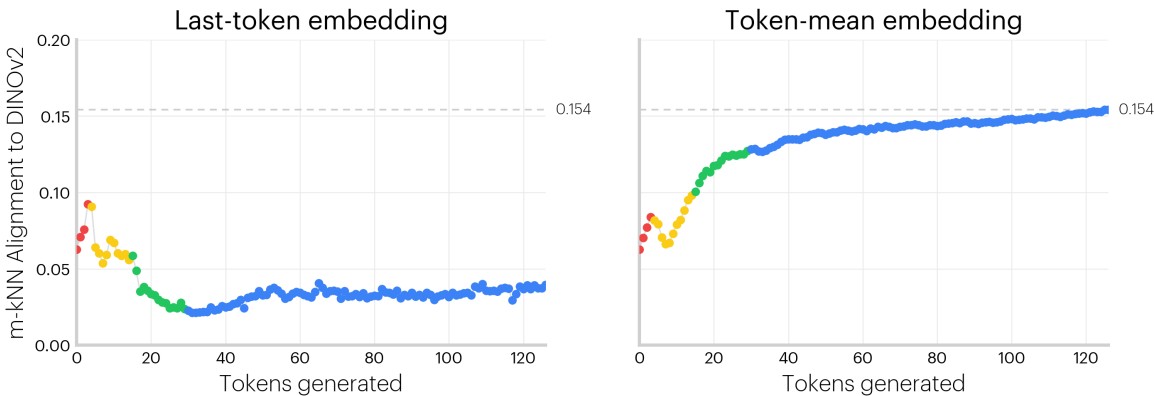

*(a)* Vision-language alignment using last-token and mean-token representations, quantified by mutual $k$-nearest neighbors. Curves are averaged over five random seeds; variability across seeds is low (mean standard deviation $2.3 \times 10^{-3}$, max $5.7 \times 10^{-3}$), hence not visible in the figure.

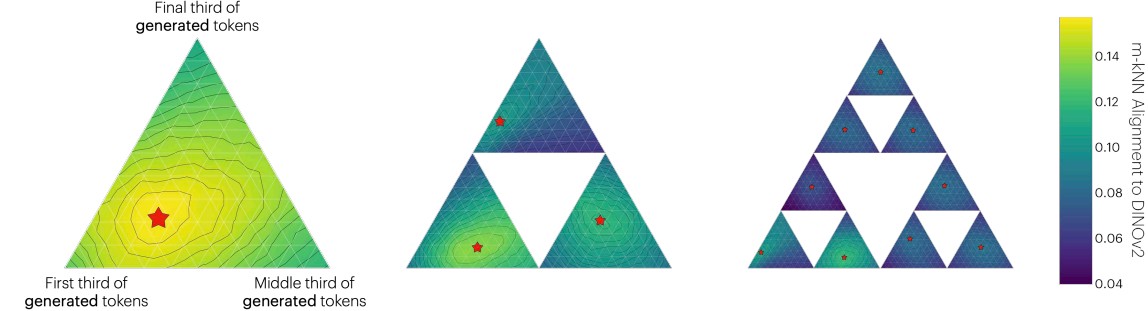

*(b)* Vision-language alignment under convex combinations of token-slice ensembles at increasing levels of granularity.

*Figure 13.* Extension of Figure 3 and Figure 4 to m-$k$NN alignment metric with $k = 10$.

the multiset of token representations but destroying their original temporal order. This removes the phase structure over token index and converts prefix averaging into averaging over randomly ordered tokens, demonstrating that the monotonic prefix trend depends on token order rather than on averaging alone. In the middle row, generated tokens are shuffled *before* re-embedding, altering the autoregressive computation itself. Alignment is reduced but remains above chance, indicating that semantic information is distributed across many tokens and is not entirely dependent on strict ordering. In the bottom row, image–text pairings are shuffled during alignment, breaking semantic correspondence across samples. In this case, alignment collapses to that expected between random kernels, confirming that the observed effects depend on meaningful cross-sample correspondence.

**Dependence on semantic correspondence.**   Figures 15 and 16 extend the convex-combination analysis to settings in which image–text pairings are shuffled. Under both debiased CKA and m-$k$NN, the interior simplex maxima observed in the main text disappear when correspondence is broken. This indicates that the convex structure is not a generic consequence of mixing representations but instead depends on shared semantic structure between text and reference embeddings.

**Different pooling methods.**   Given token-level representations $\{h_t\}_{t=1}^{T}$, we consider several pooling strategies over generated tokens:

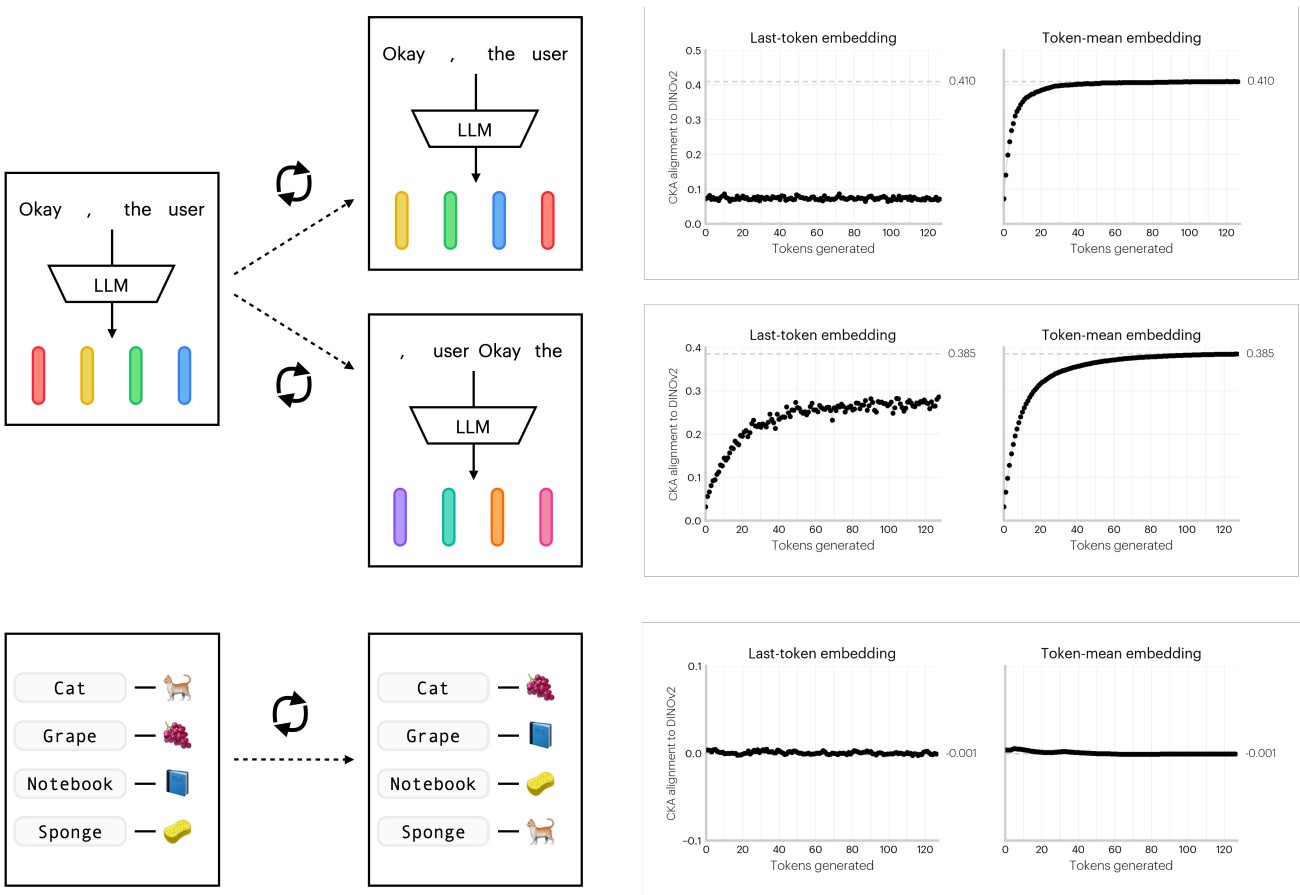

*Figure 14.* (*Top*) Token embeddings are shuffled after embedding. (*Middle*) Generated tokens are shuffled before re-embedding. (*Bottom*) Sample pairings are shuffled during kernel alignment.

$$\text{Mean pooling:} \quad \bar{h}_{\text{mean}} = \frac{1}{T} \sum_{t=1}^{T} h_t$$

$$\text{Max pooling:} \quad \bar{h}_{\text{max}}[i] = \max_{t \in \{1,\ldots,T\}} h_t[i]$$

$$\text{Best per-token:} \quad \bar{h}_{\text{best}} = h_{t^*}, \quad t^* = \arg\max_{t} \text{Align}(h_t, r)$$

$$\text{Attention pooling:} \quad \bar{h}_{\text{attn}} = \sum_{t=1}^{T} \alpha_t h_t, \quad \sum_{t=1}^{T} \alpha_t = 1$$

where $r$ denotes the reference representation and $\alpha_t$ are attention-derived weights.

As shown in Table 2, mean pooling across generated tokens achieves the highest alignment under both CKA and m-kNN, outperforming attention-based, max, and single-token representations. This suggests that semantic information is distributed across tokens, and that uniform aggregation provides a more faithful summary of the generation than selecting or reweighting individual token representations.

**Structure-based evaluation.** We evaluate mean pooling using structure-based retrieval, ranking, and clustering metrics over WiT using Qwen3-14B and DINOv2 embeddings. As shown in Tables 3–5, generation mean pooling consistently outperforms all alternatives across metrics. In retrieval, it achieves the highest overlap with vision-space neighborhoods (R@1/5/10), indicating that it best preserves local similarity structure. This trend follows in ranking metrics, where generation

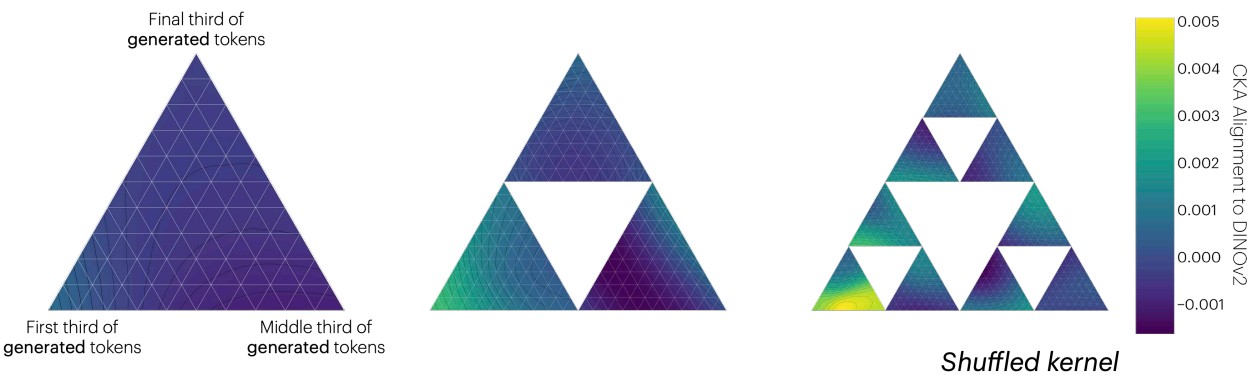

*Figure 15.* Extension of Figure 4 in which image-text pairings are shuffled, breaking semantic correspondence.

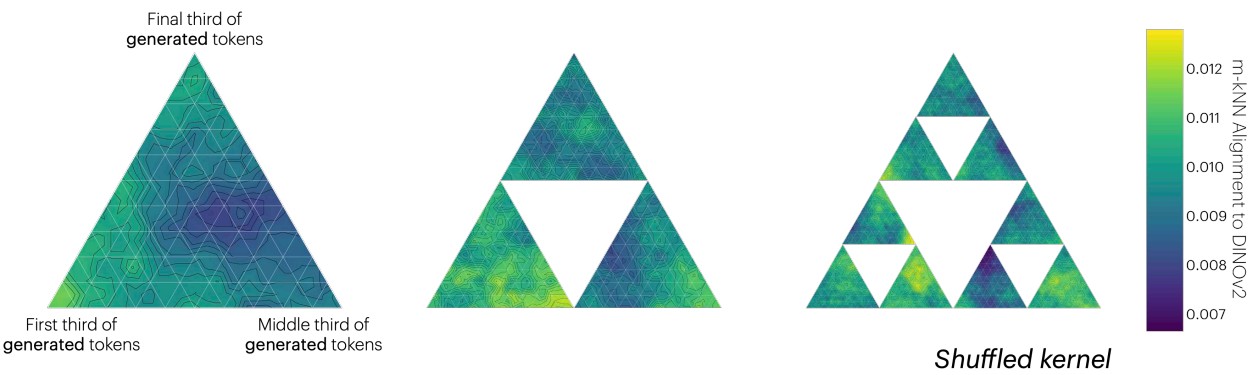

*Figure 16.* Extension of Figure 13b in which image-text pairings are shuffled, breaking semantic correspondence.

mean pooling yields higher MRR and lower mean and median rank, showing that semantically corresponding vision examples are ranked earlier. Finally, in clustering, it achieves higher ARI and NMI, indicating stronger agreement between language- and vision-space partitions. Across all evaluations, pooling over generated tokens produces representations that more faithfully recover the underlying structure than both caption-based embeddings and single-token representations.

**Geometry of token trajectories.** Figure 17 visualizes the trajectory of token-level representations for a single generation using PCA, with points colored by dataset-level alignment to the reference space. Last-token embeddings exhibit greater dispersion, while token-mean embeddings occupy a more compact region associated with higher alignment. This visualization provides geometric intuition for why averaging across tokens improves alignment: pooling moves the representation toward a stable region of representation space that is more consistent with the reference embedding.

**Language model robustness.** Figure 18 extends the tokenwise alignment analysis to additional language models and to generations of up to 512 tokens. Across models, mean pooling over generated tokens consistently outperforms individual token representations, and alignment improves as additional tokens are incorporated. Figure 19 similarly extends the convex-combination analysis, showing that interior simplex maxima persist across models. These results indicate that the observed phenomena are not specific to a single architecture or scale.

**Dependence on language model embedding space.** Figure 20 compares tokenwise alignment when generations are embedded using different models. Between Qwen3-8B and Qwen3-14B, the alignment curves and phase structure are very similar, with the main difference being a shift in absolute alignment values. This is expected, as the higher-capacity model produces embeddings that are better aligned overall, while preserving the same qualitative token-wise structure.

In contrast, when Qwen3-14B generations are embedded using gpt-oss-20B, the token-wise curves differ more substantially, and the alignment observed under the gpt-oss-20B embeddings does not strictly increase through averaging. One plausible explanation is that gpt-oss-20B is a stronger model, and the internal representations it assigns to the same text differ from

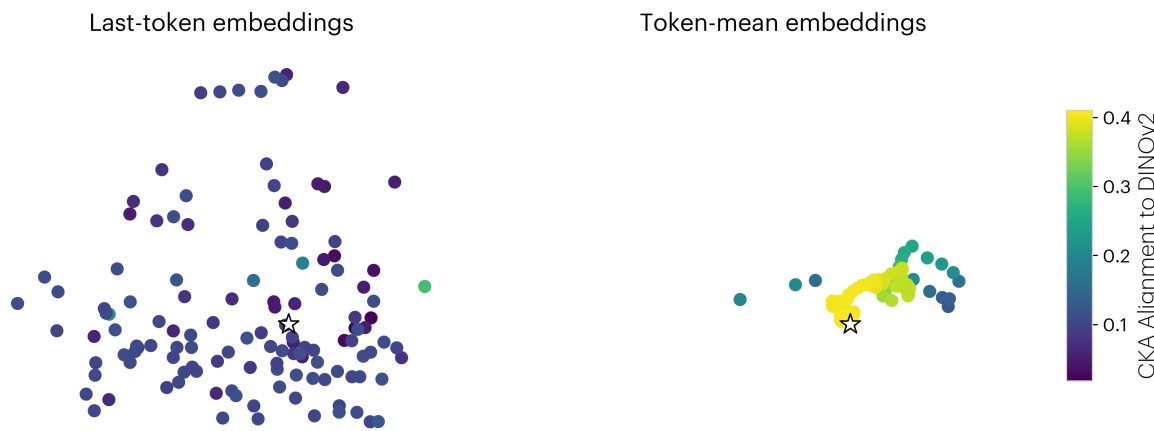

*Figure 17.* PCA of last-token and token-mean embeddings of generated tokens for a single sample from the setting of Figure 3, with colors indicating dataset-level CKA alignment to DINOv2.

those produced during generation. This mirrors the behavior observed in the main text when Qwen embeddings are applied to OLMo-generated text.

**Vision model robustness.** Figure 22 shows that using the DINOv2 [CLS] token does not change the result that mean pooling over generated tokens yields higher alignment than any individual token. Figure 23 extends tokenwise alignment analysis to additional vision models: DINOv3 (Siméoni et al., 2025), ViT-MAE (He et al., 2022), and ViT-MSN (Assran et al., 2022).

**Correctness and depth effects.** Figure 21 extends the convex-combination analysis used in reasoning tasks to additional depths of token segmentation. As the token sequence is subdivided into finer-grained segments, alignment remains maximized at interior combinations, demonstrating that the effect is stable across levels of granularity and is not driven by a particular choice of token partition.

**Representation norm and alignment.** Finally, Figure 24 examines the relationship between representation norm and alignment. Spikes in per-token and per-layer alignment are correlated with increases in representation norm. However, high norm alone does not explain the convex structure observed under token averaging and convex combinations.

# D. Layerwise

We examine whether the alignment improvements observed under token averaging also arise when representations are combined across network depth. We use these analyses as controls to distinguish effects specific to token pooling during generation from those due to generic aggregation. Figure 25 extends layerwise alignment to additional models. Across models, alignment increases with depth and peaks in later layers. Averaging across layers yields alignment comparable to the best single layer, but does not consistently improve upon it. Figure 26 evaluates alignment under convex combinations of layer-slice representations. Unlike the token-slice case, alignment is not maximized at interior simplex points and is typically dominated by later layers.

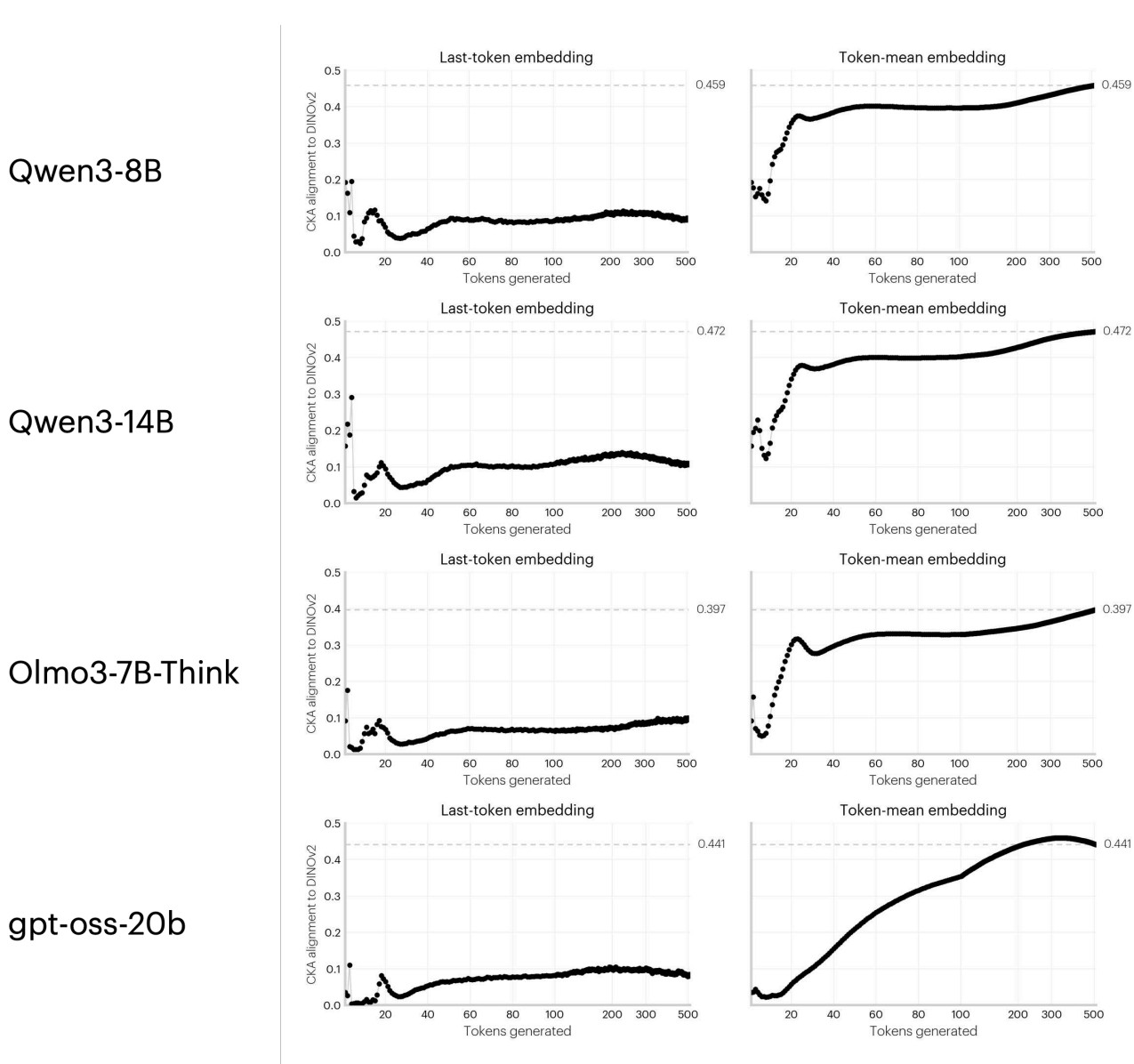

*Figure 18.* Extension of Figure 3 to additional models, using up to 512 generated tokens.

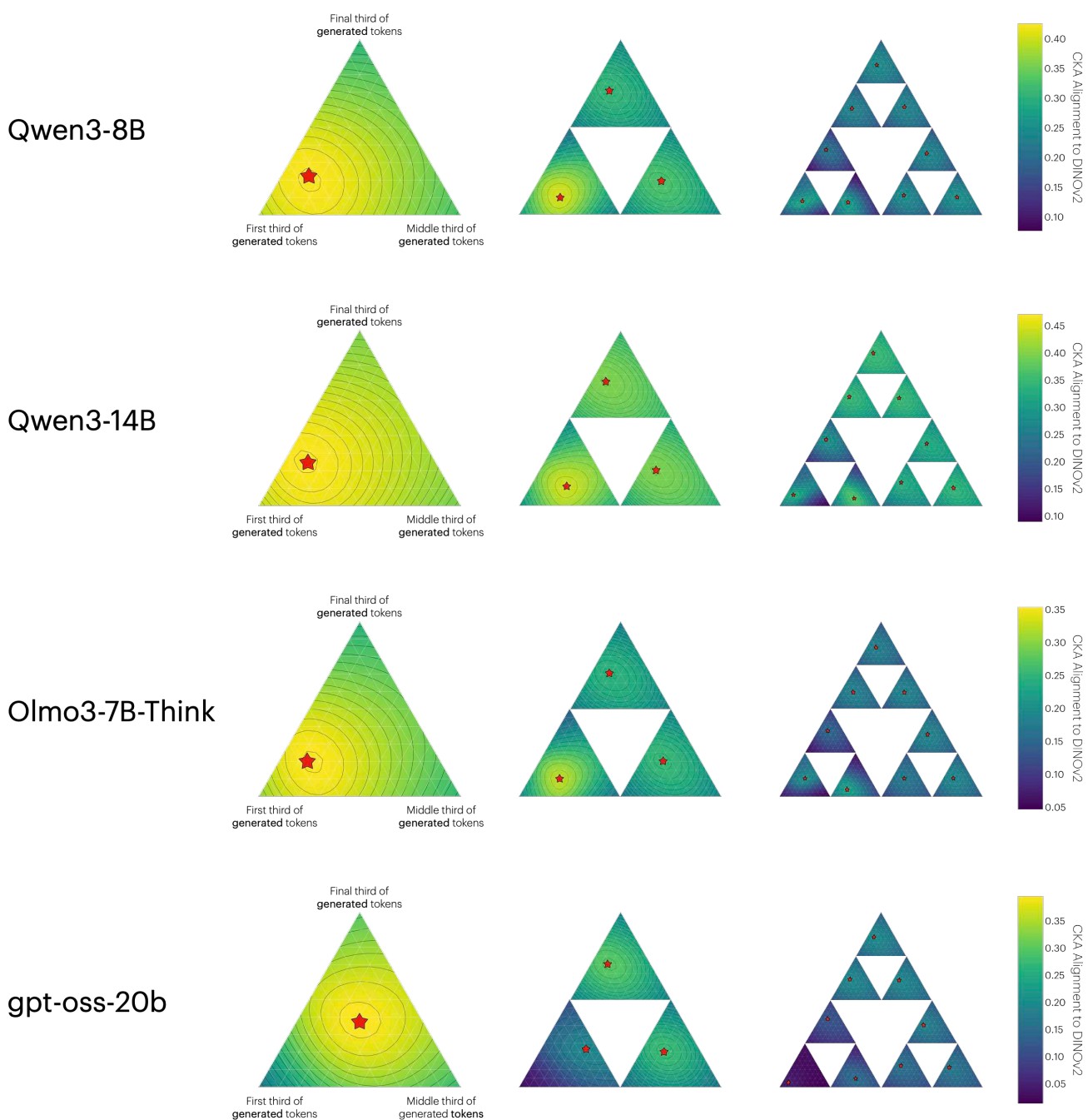

*Figure 19.* Extension of Figure 4 to additional models.

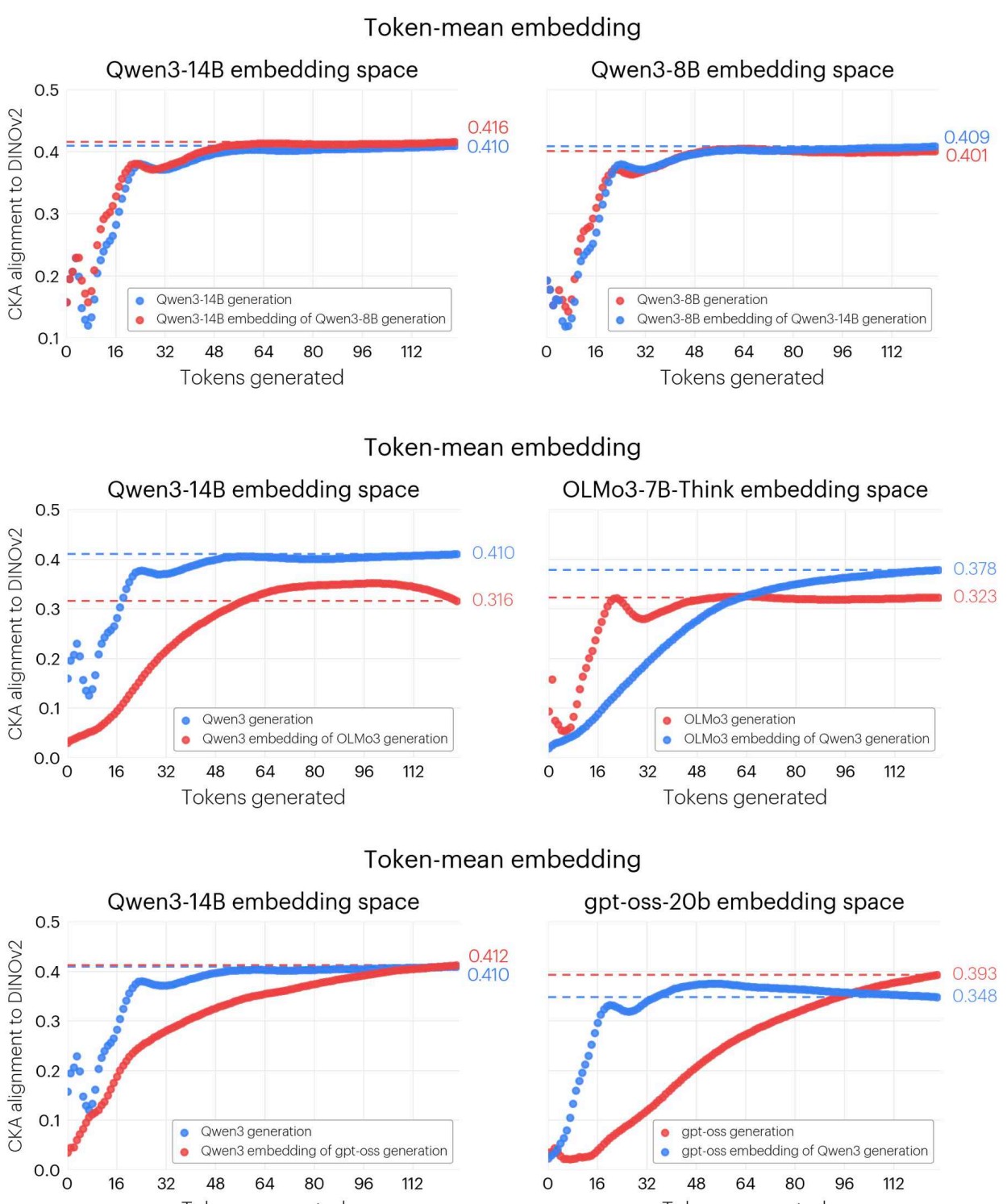

*Figure 20.* Extension of Figure 10 to additional models.

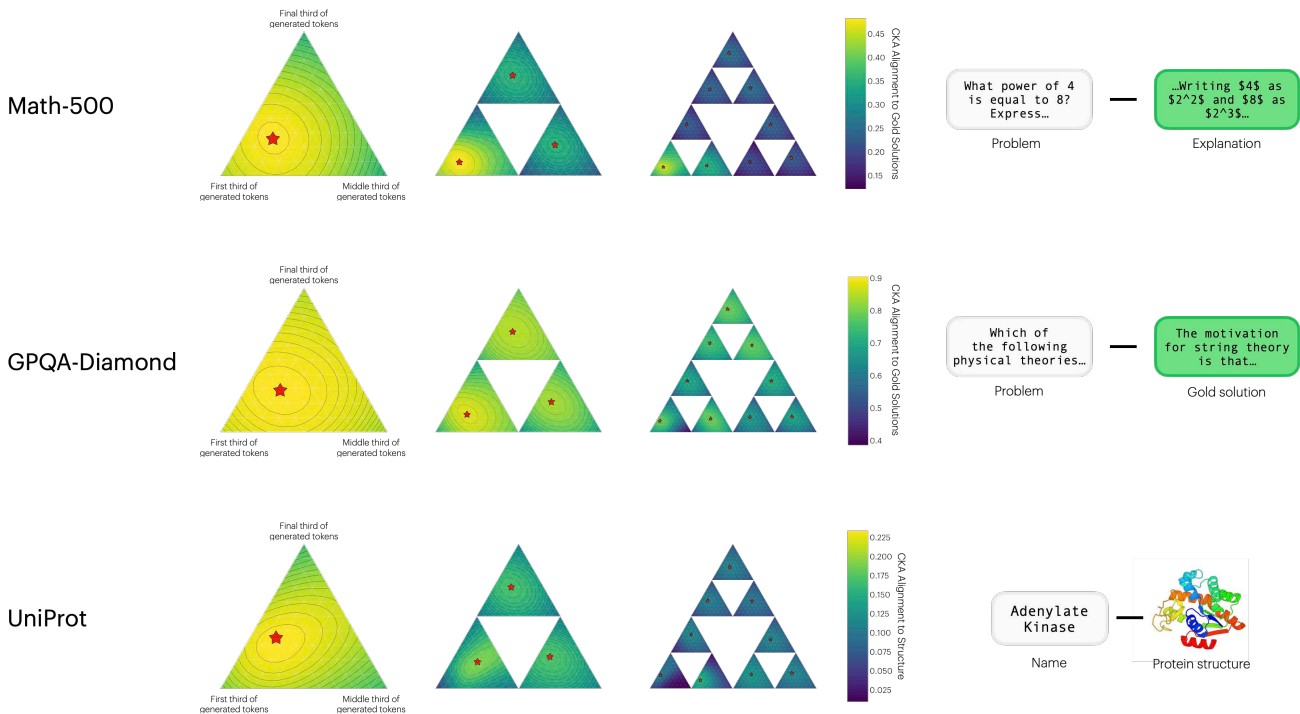

*Figure 21.* Extension of Figure 5 to additional depths.

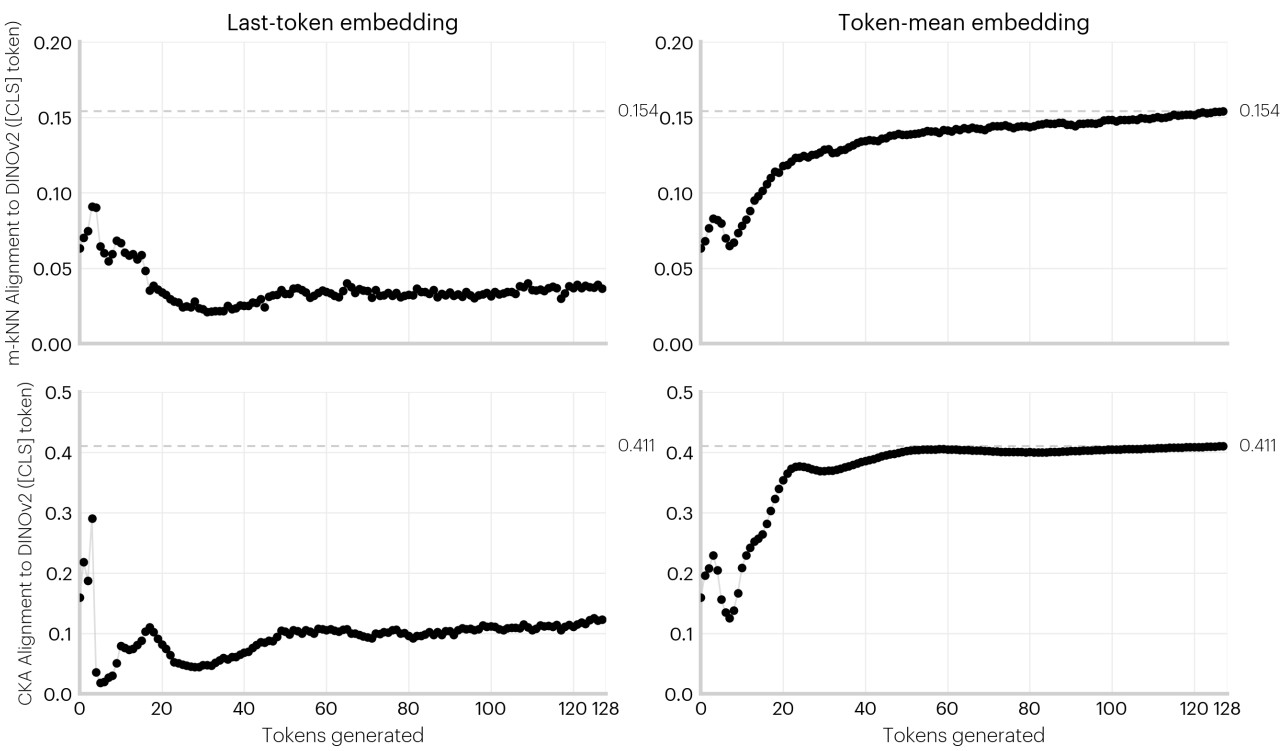

*Figure 22.* Extension of Figure 3 and Figure 13a to using the DINOv2 `[CLS]` embedding instead of mean-pooled embeddings.

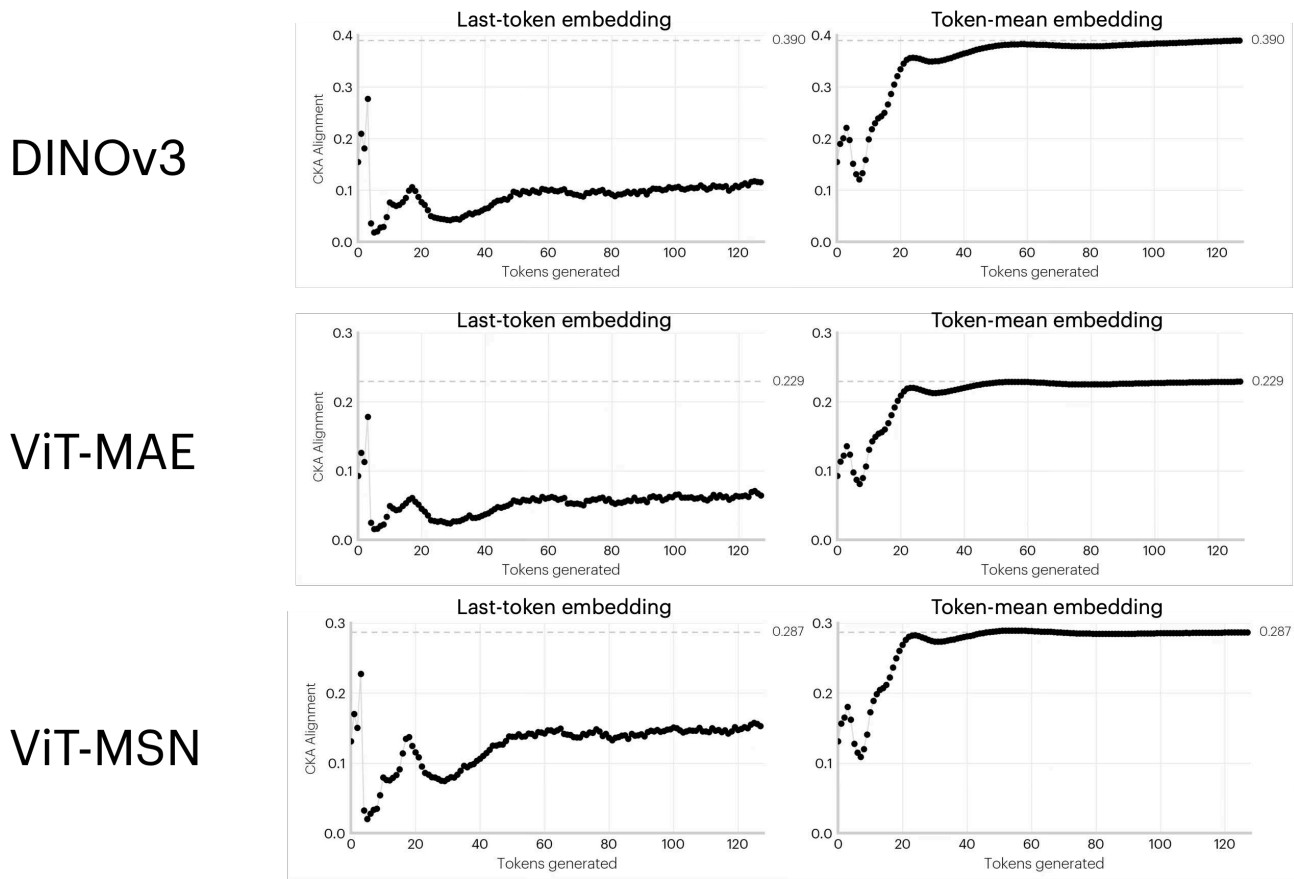

*Figure 23.* Extension of Figure 3 and Figure 13a to additional vision encoders.

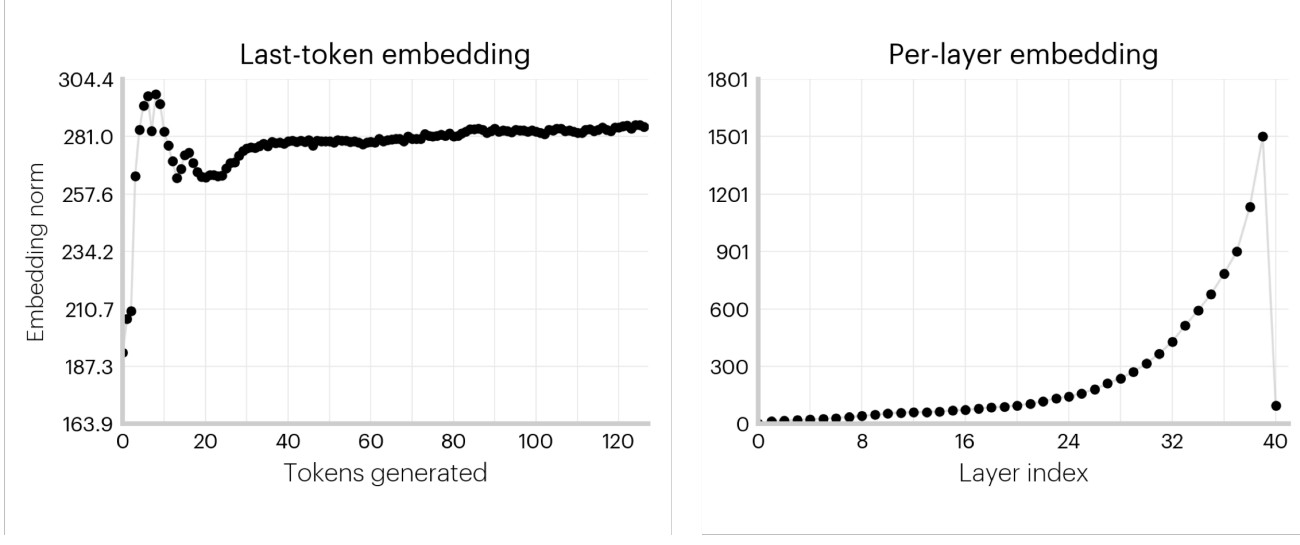

*Figure 24.* High norm correlates with spikes in per-token and per-layer representational alignment.

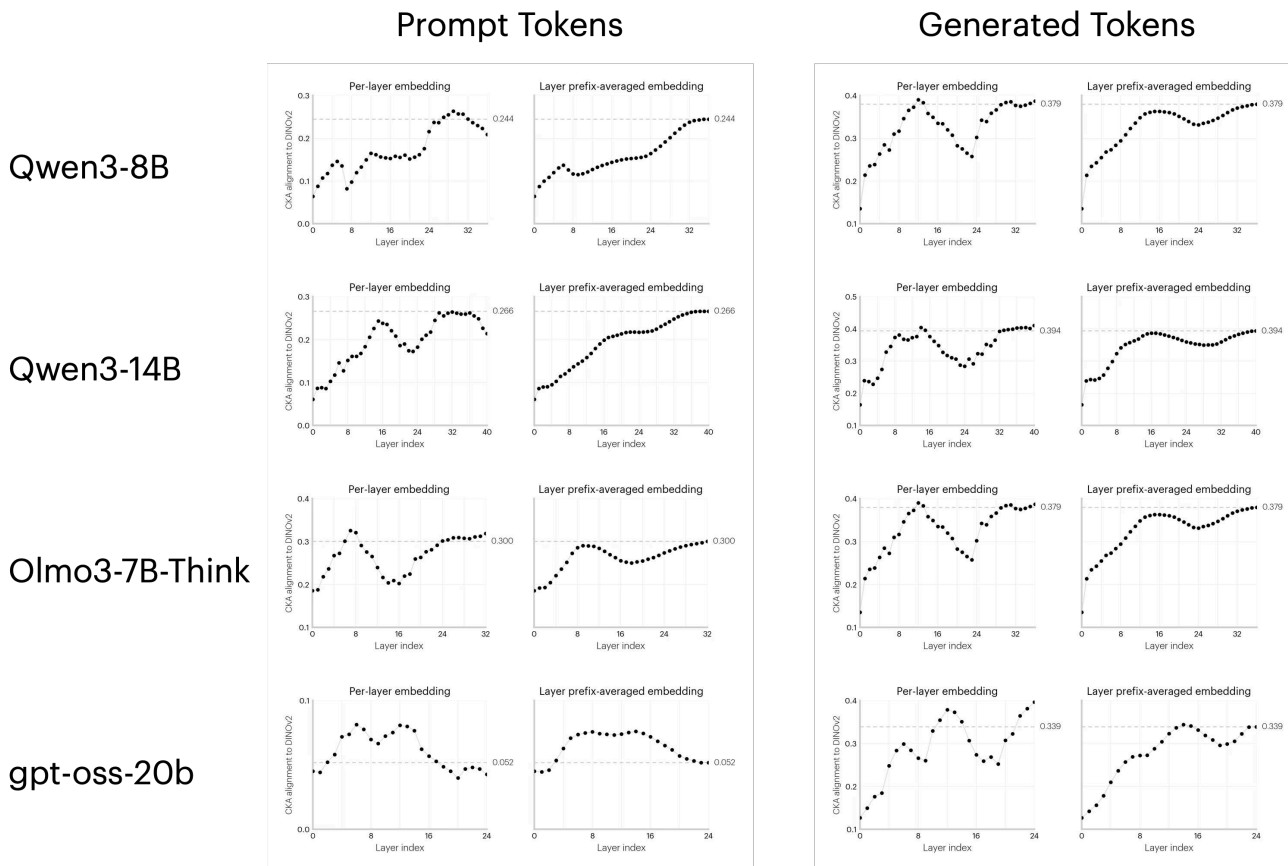

*Figure 25.* Extension of Figure 11 to additional models.

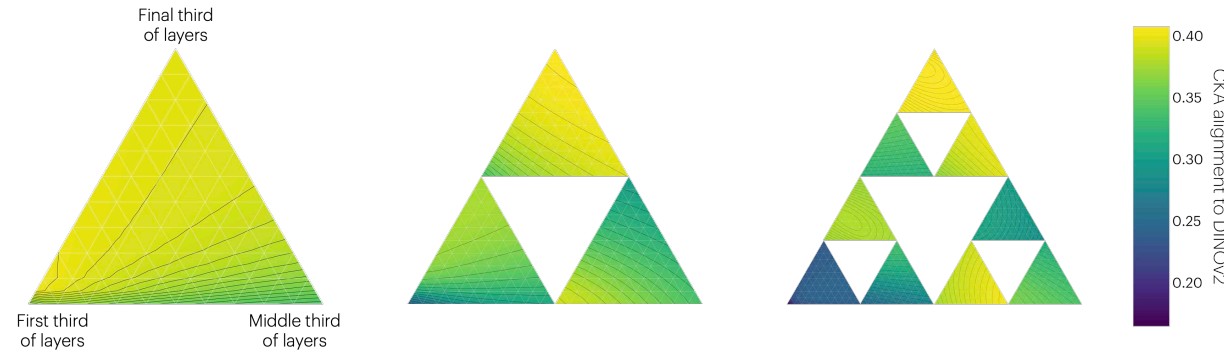

*Figure 26.* Extension of Figure 4 to vision-language alignment under convex combination of layer-slice ensembles at increasing levels of granularity.

