# OpenReview forum: "The Truth Lies Somewhere in the Middle (of the Generated Tokens)"
_ICML.cc/2026/Conference — ICML 2026 regular_

### Official Review · Reviewer_bUs4 · 2026-03-10

**Soundness:** 2
**Presentation:** 3
**Significance:** 2
**Originality:** 3
**Overall Recommendation:** 4
**Confidence:** 2

**Summary:**

The paper studies the sequence of hidden states produced during autoregressive generation. It evaluates how representations built from generated tokens align with fixed reference embeddings across vision-language, protein, and reasoning tasks. The main finding is that pooled representations over multiple generated tokens align better with these references than single-token representations.

**Compliance With Llm Reviewing Policy:**

Affirmed.

**Final Justification:**

The rebuttal addressed my main concerns in a constructive and convincing way, particularly by clarifying the contribution and refining the interpretation of key concepts. I also appreciate the authors’ commitment to improving the presentation and providing additional justification for their methodological choices.

Although some concerns regarding clarity and significance remain, I believe the paper offers an interesting and useful perspective. I therefore update my recommendation to weak accept.

**Key Questions For Authors:**

What is the scientific contribution of this paper?

**Limitations:**

yes, but it was a little vague. I would like the authors to be more specific.

**Strengths And Weaknesses:**

The main empirical pattern is interesting and the paper is written in a clear language. The results suggest that token-mean representations outperform individual tokens and that mixtures of tokens often outperform any single segment alone.

My main concern is that the paper’s framing and terminology are often not introduced clearly enough for readers outside the immediate subarea. For example, Section 2.1 says “we use hte unbiased centered kernel alignment (CKA) metric,” which is both a typo and too small for an introduction for a metric that is central to the entire analysis. I am missing a motivation for this metric. Another clarity problem concerns the term “grounding.” The paper states, “We use grounding to denote correspondence between a model’s internal representation and an external structure that constrains correctness independently of text.” Here I was wondering if it is the methodology of comparison or if grounding is something more specific serving the comparison that was not explained further. Later, when you write something like “more grounded” I was confused if this now means better aligned. I think it does after reading the full paper and knowing the context, but it would be better to make it clear from the first usage.

Similarly, I think the paper should explain some terms more carefully: “external structure,” what a “correctness-relevant structure” is, and what is meant by “external sensory modality.” As the reader I had to stop and think about what could have been meant with that, make an educated guess and read on. If there is no space in the paper, a reference to an appendix section explaining would be fine, but that would also interrupt the reading flow.

In 2.8 I think I am understanding a bit but not fully: You are using the decoder to obtain the representations and compare against encoder generated output because you want to compare against embeddings? Adding one or two sentences for a detailed description would help not having to wonder about that.

In 2.8 and 2.9 the authors present the models and data that produce the references. I would have welcomed an explanation why these specific models and data are fit for the task. I am also questioning the “correctness” of representations, if they as well stem from a model. Is this standard practice? In the field I am from it would be considered a gross error to label anything stemming from a model as correct, especially as ground truth, with this definiteness. I think this issue is described in the first paragraph in the limitations but I am not completely sure. It is a bit vague there and before quite explicit.

While I really liked Figure 3 and 4 that helped me understand the findings, I had a little trouble understanding Figure 1. The paper says, “As illustrated in Figure 1, we find that pooling across generated tokens yields representations that are more grounded than any individual token,” but I am not sure how to find this comparison of pooling and individual tokens in Figure 1. Maybe the difference between the two can be highlighted more clearly. As it stands, the wording overstates what Figure 1 itself shows.

For clearness I would have liked the three domains to be presented more clearly from the beginning on. The application to protein-language is not really properly introduced and I stumbled over it in the text. I would suggest naming the three domains in the first half page of the paper, either in the abstract or the beginning of the introduction.

Beyond clarity, I still have the broader concern that the significance of the contribution is somewhat hard to judge. The authors are not proposing a method that improves training, inference, or downstream performance; instead, they propose a way to analyze internal representations during generation. The paper’s more specific contribution seems to be the combination of generated-token pooling, grounded-reference alignment, and convex-mixture analysis across token segments. That is a coherent package, but it feels more like a particular analysis design than a major conceptual leap, especially given how intuitive the high-level conclusion is: that information is spread across the generation process rather than concentrated in one final chunk. The value of the paper therefore depends heavily on whether one sees the reported alignment geometry as a substantial new scientific insight or as a careful quantification of something many readers might already expect.

Overall, I think this is a thoughtful paper with a real empirical observation, but I was left with substantial concerns about clarity of terminology, strength of interpretation, and significance.

---

> ### Author Rebuttal · Authors · 2026-03-30
>
> Thank you for taking the time to write a thoughtful review! We are encouraged that you find our results interesting and paper clearly written. We kindly respond to your concerns and number them for your convenience:
>
> # 1. What is the scientific contribution of this paper?
>
> In our paper, we quantitatively study a scientific observation about the generated tokens of a language model. The immediate significance of our work is that it provides a new perspective for the representation learning and interpretability community to understand model behavior. In particular, recent work has shown strong interest in cross-modal convergence and alignment (e.g., “The Platonic Representation Hypothesis”), yet these works typically study representations as static embeddings. We instead study how representations evolve during generation, and show that viewing them as dynamic across tokens and layers can reveal structure in how models process information that is not visible from a static embedding perspective. We believe this perspective is especially relevant to understanding how and when such alignment emerges.
>
> Studying representation evolution over generation is a new perspective that we introduce, and using this method, we make initial observations of behaviors that open up concrete directions for future work. For example, we observe phases such as recall, which raises the question of why the model appears to bring an internal world model into view during generation. This is a type of structure that would not be visible from a single embedding.
>
> Another example is our observation that stronger model embeddings of generations from weaker models do not exhibit the same convexity phenomenon (Fig. 9), which could help illuminate how models process information that does not match their world model. Finally, the layer-wise results provide insight into why we observe higher alignment in later layers, which is somewhat surprising given recent works finding that middle layers capture more concept-level representations (“Layer by Layer: Uncovering Hidden Representations in Language Models”), while the final layers are primarily used for task-specific decoding.
> Overall, these findings suggest that generation-time representations are a useful object of study in their own right, and that analyzing trajectories of hidden states can provide a general framework for understanding model behavior.
>
> # 2. My main concern is that the paper’s framing and terminology are often not introduced clearly enough for readers outside the immediate subarea.
>
> Regarding specific suggestions, such as clarifying the wording of Figure 1 and terminology such as “external structure” and presenting the three domains explicitly earlier in the paper, we will incorporate this into our revision.
>
> # 3. You are using the decoder to obtain the representations and compare against encoder generated output because you want to compare against embeddings?
>
> Our goal is to compare the representations produced during autoregressive generation to representations that capture structure from an external domain. To do this, we use encoder models (e.g., vision or protein encoders) that produce embeddings reflecting properties of that domain. This methodology has been standardized in recent years in representation learning literature, especially for works studying cross-modal convergence.
>
> # 4. I am also questioning the “correctness” of representations, if they as well stem from a model. Is this standard practice?
>
> Thank you for raising this point. We do not treat the reference representations as ground-truth in a strict sense. Instead, we use them as external reference spaces that capture structure relevant to the domain (e.g., visual similarity or protein structure), learned from data independent of the language model. In this sense, “correctness” refers to consistency with an external structure that is not derived from the text or the language model itself, rather than absolute ground truth. This type of comparison is common in representation learning, where independently trained encoders are used as reference spaces to evaluate alignment or transfer.
>
> We thank you for the suggestion and agree that the wording in the paper could be clearer; we will revise it to avoid implying that these representations are treated as definitive ground truth.
>
> ---
>
> Thank you again for your helpful feedback! Do these answers address your concerns with the paper? If not, what further clarification or modifications could we make to improve your score?

---

> > ### Author Rebuttal · Reviewer_bUs4 · 2026-04-03
> >
> > Thank you for the detailed and thoughtful responses. In particular, I greatly appreciate the clarifications regarding the scientific contribution. I would strongly encourage incorporating these explanations into the paper, as they would help readers outside the immediate field better understand and benefit from your work.
> > I still have some remaining questions regarding the suitability of the models introduced in Sections 2.8 and 2.9, as well as the advantages of using the CKA metric. If you could further elaborate on these points—and potentially include this discussion in the paper (e.g., in an appendix)—it would strengthen the paper. Addressing these aspects would also positively influence my evaluation.

---

> > > ### Author Response · Authors · 2026-04-03
> > >
> > > Thank you for your feedback! We agree, and will incorporate your suggestions into the revision. We address these other aspects that you mention:
> > >
> > > # 1. suitability of the models introduced in Sections 2.8 and 2.9
> > >
> > > Our main experiments quantify vision–language kernel alignment. We choose to use the Qwen3 model family in the main text because it is a capable open-source language model. However, we note that using a different model family does not affect the observed trends; we extend our main experiments to the Olmo3 and gpt-oss model families in Appendix Figure 18.
> > >
> > > On the vision side, we choose to use the DINOv2 vision encoder in the main text because it is well established in the field. As per the suggestion of reviewer sEex, we have also extended our main experiments to additional vision models to improve robustness. In particular, we extend Figure 3 (CKA) and Figure 13a (m-kNN) to three additional vision encoders: DINOv3 (base), ViTMAE (base), and ViTMSN (base). DINOv3 figure: https://imgur.com/a/rkd7q62. ViTMAE figure: https://imgur.com/a/fFdrLx8. VitMSN figure: https://imgur.com/a/Kguj3UX. We find that the main result holds, but the degree of alignment varies depending on the reference encoder.
> > >
> > > All models used are standard in the field and open-source.
> > >
> > > # 2. advantages of using the CKA metric
> > > We describe our use of the unbiased CKA metric and compare it to other metrics in Appendix A. Please let us know if this addresses your question adequately.

---

### Official Review · Reviewer_pE2B · 2026-03-12

**Soundness:** 2
**Presentation:** 4
**Significance:** 3
**Originality:** 3
**Overall Recommendation:** 4
**Confidence:** 3

**Summary:**

This manuscript proposes to extract a single representation from the sequence of hidden states produced by a model during autoregressive generations using mean pooling across the generated tokens. This representation aligns better with grounded reference embeddings, showing that semantic information is spread across the representations of the generated tokens. These embeddings have been compared to reference embeddings extracted from models that encode structural information of the content generated ( images, reasoning and proteins).

**Compliance With Llm Reviewing Policy:**

Affirmed.

**Final Justification:**

I recommend a weak accept. The authors propose extracting a single representation from the sequence of hidden states produced during autoregressive generation by applying mean pooling across the generated tokens. The authors evaluate whether pooled embeddings are semantically richer by comparing them to reference embeddings. The approach appears sound and is clearly presented. However, there are concerns regarding its originality and broader relevance to the field.
In the rebuttal, the authors addressed my main concerns, showing that mean pooling benefits downstream tasks and arguing that the results are not biased, despite the reference embeddings also being constructed via mean pooling (limited to a single task). This improved my evaluation.
However,  I do not recommend a strong accept, as mean pooling is already widely known to be effective, and the novelty here appears limited. Nonetheless, the work merits a weak accept due to its thorough empirical evaluation and clear presentation.

**Key Questions For Authors:**

1. The work evaluated three main strategies. Could you please provide a comparison with other alternatives from the literature?
2. Could you provide a clearer interpretation of the magnitude of the gains shown in the graphs? Figure 14 shows a very low score (-0.001) while some earlier experiments showed alignment scores around 0.1, and it is hard to compare.
3. Figure 18 reports that increasing context length improves the alignment score. What exactly does this imply about the representation, and how should this be interpreted?
4. Can you design a control experiment that would test whether pooled representations truly capture semantics better, rather than simply resembling the construction of the reference embeddings?
5. Even if these pooled embeddings are not suitable for re-injection into the model for generation, are they useful in downstream tasks such as retrieval, clustering, or ranking?
6. Could you please elaborate on how pooled hidden states are semantically better representations, rather than just better-aligned ones?

**Limitations:**

yes

**Strengths And Weaknesses:**

Strengths:
- The authors show that averaging the representation of the generated tokens creates a representation that aligns better to the reference embedding. The authors ruled out simpler explanations through an extensive set of ablation studies in the Appendix, and presented very interesting results.
- Using grounded reference spaces is quite appropriate to represent the semantics of the concept/element represented.

Weaknesses:
- The work is observational: while it documents robust convexity phenomena, there is little explanation of why hidden-state trajectories exhibit this geometry.
- The experiments are conducted across different domains and across three main setups: last layer token embeddings, mean embeddings, and convex combinations of the embeddings. Even though the motivation is clear, it is hard to compare different domains, when the embeddings were constructed differently. For instance, whenever a language model is used, the references match with the setup tested, while they remain the same for other models.  Furthermore, a major methodological concern is that some of the reference embeddings constructed could be biased towards pooled representations. If the evaluation target is itself an average representation, it would make sense that performing a similar operation to create reference embeddings would increase alignment. Furthermore, claims are very strong given that the authors do not report the performance of the selected reference models selected.
- Given the paper’s claims, it would be important to compare the setups proposed against established alternatives in the literature (ex: attention-based pooling, max pooling, pooling over a selected subset, etc.).
The graphs are difficult to interpret because the paper does not make clear how large, meaningful, or statistically significant the reported gains from pooled embeddings are.
- Many design choices (such as token cutoff, three slices) are fixed with little explanation. For example, the results suggest that increasing the context length improves the alignment, but the implications of this finding are not discussed.
- Even though the authors acknowledged that averaging hidden states across tokens does not yield an embedding that can be easily fed back into the model to start or continue generation, the evaluation of the embeddings is strictly at the representational level (there is no experiment on how these embeddings improve downstream tasks, which makes the practical aspect of the study very limited).
- Typos in lines 081 (the) and 406 (scene-relevant)

---

> ### Author Rebuttal · Authors · 2026-03-30
>
> Thank you for taking the time to write a thoughtful review! We are encouraged that you find our results interesting and ablation studies extensive. We kindly respond to your concerns and number them for your convenience:
>
> # 1. Could you please provide a comparison with other alternatives from the literature?
>
> We will incorporate your suggestion into the revision by explicitly comparing to standard pooling strategies from the literature. In addition to mean pooling and per-token baselines, we evaluate max pooling and attention pooling over generated tokens. Figure: https://imgur.com/a/ChE3yvj.
>
> Empirically, mean pooling outperforms both alternatives. This suggests that the gains we observe are not due to pooling in general, but to averaging across tokens, which preserves complementary information distributed over the generation.
>
> # 2. Could you provide a clearer interpretation of the magnitude of the gains shown in the graphs?
>
> We will incorporate your suggestion into the revision by clarifying the magnitude of the gains. The relative improvements vary by metric (e.g., CKA vs. m-kNN), and we will make this more explicit in the paper. For CKA in particular, we will contextualize the observed changes using prior work (e.g., [1]) to clarify how differences in this range should be interpreted. Since similarity metrics are less directly interpretable than downstream task metrics, we also include additional downstream experiments (see response to 5.) that provide more interpretable measures of improvement.
>
> # 3. Figure 18 reports that increasing context length improves the alignment score. What exactly does this imply about the representation, and how should this be interpreted?
>
> Thank you for the question! The interpretation of context length improving alignment is that the representations are progressively refined during autoregressive generation. Longer generations allow the model to produce representations that are more consistent with the reference encoder as it follows the prompt instruction to elaborate on the caption (see sample generations in Appendix B.2).
>
> # 4. Even if these pooled embeddings are not suitable for re-injection into the model for generation, are they useful in downstream tasks such as retrieval, clustering, or ranking?
>
> Thank you for the suggestion. We include this result in our revision by evaluating pooled representations on downstream tasks. We extend the setting of Figure 3 by comparing generated-token mean pooling against prompt-side mean pooling, prompt last-token pooling, and generation last-token pooling on tasks that assess whether representations preserve the semantic structure of the vision embedding space. Figure: https://imgur.com/a/v1eEc7q. Generated-token mean pooling performs best across all metrics, indicating that pooled representations are not only better aligned but also more useful for downstream tasks such as retrieval, ranking, and clustering.
>
> # 5. Could you please elaborate on how pooled hidden states are semantically better representations, rather than just better-aligned ones?
>
> The response to 4. shows that an implication of higher alignment is better performance on semantics-based downstream tasks. Additionally, we would like to clarify that in the representation learning community it is standard practice to quantify representational alignment as a proxy for semantic usefulness, particularly in work on cross-modal convergence and alignment (e.g., [2]). Alignment is itself a meaningful objective that reflects how well representations capture shared relational structure, and is widely used to study and compare representations across models and modalities. For example, there are works interested in the task of explicitly aligning representations [3].
>
> In our paper, we quantitatively study a scientific observation about the generated tokens of a language model. The immediate significance of our work is that it provides a new perspective for the representation learning/interpretability communities to understand model behavior. In particular, prior work treats representations as static embeddings, whereas we study how representations evolve during generation, and show that viewing them as dynamic across tokens and layers can reveal structure that is not visible from a single embedding.
>
> Using this perspective, we observe behaviors such as recall, which suggests that the model appears to bring an internal world model into view during generation. This would not be observable from a single embedding, and supports the interpretation that pooled generated representations reflect meaningful semantic structure.
>
> ---
>
> Thank you again for your helpful feedback! Do these answers address your concerns with the paper? If not, what further clarification or modifications could we make to improve your score?
>
> - [1] - “Similarity of Neural Network Representations Revisited”
> - [2] - “The Platonic Representation Hypothesis”
> - [3] - “Harnessing the Universal Geometry of Embeddings”

---

> > ### Author Rebuttal · Reviewer_pE2B · 2026-04-02
> >
> > We thank the authors for the clarifications. The authors have successfully addressed most of our concerns. The comparisons with prior methods and the evaluation on downstream tasks provide evidence that the pooled representations are useful and outperform the considered alternatives.
> >
> > However, several points raised in the weaknesses remain unclear:
> > 1. Question 4 remains unaddressed. Can the authors design an experiment to demonstrate that the reference embeddings are not biased toward pooled representations, given that they are also constructed via average pooling?
> > 2. Can the authors further justify their design choices, such as the token cutoff and the use of three slices?

---

> > > ### Author Response · Authors · 2026-04-02
> > >
> > > Thank you for your response! We appreciate your helpful suggestions and further address your concerns:
> > >
> > > # 1. Can the authors design an experiment to demonstrate that the reference embeddings are not biased toward pooled representations, given that they are also constructed via average pooling?
> > >
> > > We appreciate the suggestion and will discuss this in the revision. For the main experiments that use vision embedding, we agree that we could use the CLS token embedding. We replicate Figure 3 (CKA) and Figure 13a (m-kNN) and note that the observed trend still holds. Figure: https://imgur.com/a/FLA2TXO. We will also include figures for results involving different pooling methods for language reference embeddings in the revision.
> > >
> > > # 2. Can the authors further justify their design choices, such as the token cutoff and the use of three slices?
> > >
> > > ***Token cutoff***. We clarify that our observations from Figure 3 (CKA) and Figure 13a (m-kNN) hold through 2048 tokens, which will be included in the revision. Figure: https://imgur.com/a/48oiBLv. Alignment saturates after ~512 tokens. The choice of 128 tokens in Figure 3 and Figure 13a is to more clearly show the shape of the early tokens while keeping a linear x-axis scaling, and we will clarify this in the revision.
> > >
> > > ***Three slices***. Our goal is to test whether mixing generated token representations improves alignment even when the contributing tokens are not individually optimal. In principle, this requires evaluating every subset of tokens, however the number of possible subsets for 128 tokens is $\sum_{k=1}^{128} \binom{128}{k} = 2^{128}-1$, which is intractable to evaluate. Instead, we approximate this space using structured mixtures over contiguous slices. Partitioning the sequence into three segments allows us to (1) mix information from distinct phases of generation and (2) study convex combinations over these regions in a tractable, low-dimensional setting. This provides a more simplified experiment of token mixing while still addressing our hypothesis that combining representations from different parts of the sequence improves alignment. To further validate that this choice is not an artifact of coarse partitioning, we refine the approximation through recursive subdivision (e.g., 9 and 27 slices). As shown in Figure 4, the same convexity structure persists at finer granularities, implying that the effect is not specific to the three-slice setting.
> > >
> > > ---
> > >
> > > Let us know if this addresses your comments!

---

### Official Review · Reviewer_sEex · 2026-03-12

**Soundness:** 3
**Presentation:** 3
**Significance:** 3
**Originality:** 3
**Overall Recommendation:** 4
**Confidence:** 4

**Summary:**

The paper looks at how to compress the sequence of hidden states produced by an autoregressive language model during generation into one single representation. Rather than focusing only on embeddings from the input prompt, the authors study the hidden states of generated tokens and compare individual token representations with pooled ones, such as averaging, last-token representations, and more general convex combinations. These are evaluated against fixed reference embedding spaces from different domains, including vision-language data, reasoning tasks, and protein representations, using unbiased CKA as the main similarity metric.

The main result is quite consistent: averaging over generated tokens gives representations that align better with the reference embeddings than any single token does on its own. The paper also finds a fairly stable token-level “phase structure” during generation, and shows that convex combinations across different parts of the generated sequence often reach their best alignment inside the simplex, not at the corners, which suggests the useful information is distributed rather than concentrated in one place.

**Compliance With Llm Reviewing Policy:**

Affirmed.

**Final Justification:**

I remain at Weak Accept after reading the rebuttal.

The additional experiments on longer contexts, shuffled-token controls, alternative reference models, statistical testing, and downstream tasks substantially strengthen the paper and address many of my earlier concerns. These results increase my confidence that the main empirical finding is robust and potentially useful.

My remaining concerns are about interpretation and impact. The paper demonstrates improved alignment to external reference spaces, but stronger claims about grounding or semantic faithfulness should be stated more carefully. In addition, the contribution is primarily an empirical and analytical one, built around a relatively simple aggregation strategy, so I do not currently see enough for a stronger recommendation.

Overall, I think the paper merits acceptance as a solid contribution, but at the level of Weak Accept rather than a stronger accept.

**Key Questions For Authors:**

-What happens to consistency as you push the generation length well beyond 128/512 tokens — say, to 1024, or 2048? Does consistency keep increasing monotonically, or does it eventually level off / start to decline? This seems pretty critical for understanding the actual limits of where this observation applies.

-Could you suggest a stronger test for teasing apart the “distributed semantics” hypothesis from the “variance reduction” one?

-How do the results change if you switch to a different reference model? If the convex structure depends on the particular reference model being used, that would weaken the interpretation of the result as an intrinsic property of the generative model.

**Limitations:**

The authors discuss the main limitations: the indirectness of the metric, the fact that averaged representations cannot be used in generation.
However, a few other limitations are not addressed:

-Dependence on specific reference models.

-No statistical significance tests are reported.

-A fixed temperature / decoding strategy is used, and its impact is not explored.

**Strengths And Weaknesses:**

Strengths:
1) Soundness:
The paper asks a clear and meaningful question: if we want one representation from generation, which hidden states should we aggregate? Its main empirical result that mean pooling over generated tokens often works better than the best single token  looks fairly robust across several domains, models, and control settings.

2) Presentation:
The overall story is easy to follow: the paper sets up the aggregation problem, motivates agreement-based evaluation, and then shows that pooled generation states tend to outperform single-token states. The manuscript also stays mostly focused on one central idea, which helps.

3) Significance:
This is a relevant question for interpretability and representation learning, especially for decoder-only LMs in actual generation settings. The broader idea of looking at hidden-state trajectories rather than isolated token states feels useful and potentially important.

4) Originality:
The novelty is mostly in the perspective rather than in new machinery, but that perspective does feel fresh. Focusing on generated-token trajectories instead of standard fixed-input embeddings is an interesting and memorable angle.

Weaknesses:
1) Soundness:
My main concern is that the paper seems to support a weaker claim than it sometimes states: it shows better alignment to reference spaces, but not necessarily that pooled states reflect the input more faithfully or encode semantics in a complementary way. Since the evaluation relies on indirect proxies like CKA, and some reference spaces are themselves debatable, the stronger interpretation feels a bit under-justified.

2) Presentation:
The wording is occasionally stronger than the evidence really supports — terms like “grounding,” “correctness,” or “reflect the input more accurately” sound slightly overstated.

3) Significance:
Right now the contribution feels more descriptive than actionable. The paper does not yet show clear downstream benefits, so the practical impact is still somewhat speculative.

4) Originality:
Most of the technical ingredients are standard, so the contribution really rests on the empirical phenomenon and its framing. For the paper to feel fully convincing, it probably needs either a stronger explanation of why the effect happens or a clearer demonstration of practical payoff.

---

> ### Author Rebuttal · Authors · 2026-03-30
>
> Thank you for taking the time to write a thoughtful review! We are encouraged that you find our research question meaningful and paper easy to follow. We kindly respond to your concerns and number them for your convenience:
>
> # 1. What happens to consistency as you push the generation length well beyond 128/512 tokens — say, to 1024, or 2048?
>
> As per your suggestion, we extend Figure 3 (CKA) and Figure 13a (m-kNN) to 2048 tokens, which will be included in the revision. Figure: https://imgur.com/a/48oiBLv. Alignment saturates after ~512 tokens.
>
> # 2. Could you suggest a stronger test for teasing apart the “distributed semantics” hypothesis from the “variance reduction” one?
>
> We appreciate the suggestion, and will incorporate it into our revision. A stronger test is to fix the number of averaged vectors while changing whether they come from the same generation trajectory. We compare averaging the model’s true generated tokens to averaging token embeddings that are increasingly shuffled across samples, which keeps the amount of averaging the same but breaks semantic consistency. If the gains were mainly due to variance reduction, these controls should perform similarly. Instead, alignment decreases steadily as more tokens are shuffled: the true generated mean achieves m-kNN $= 0.1426$ and CKA = $0.3593$, compared to $0.1378$/$0.3540$ at 25% shuffling, $0.1307$/$0.3389$ at 50%, $0.0956$/$0.2670$ at 75%, and $0.0094$/$0.00058$ at 100%. This suggests that the benefit of averaging comes from combining semantically consistent token representations, not just reducing variance.
>
> # 3. How do the results change if you switch to a different reference model?
>
> As per your suggestion, we extend Figure 3 (CKA) and Figure 13a (m-kNN) to three additional vision encoders: DINOv3 (base), ViTMAE (base), and ViTMSN (base). DINOv3 figure: https://imgur.com/a/rkd7q62. ViTMAE figure: https://imgur.com/a/fFdrLx8. VitMSN figure: https://imgur.com/a/Kguj3UX. We find that the main result holds, but the degree of alignment varies depending on the reference encoder.
>
> # 4. No statistical significance tests are reported.
>
> We will incorporate your suggestion into the revision. As an initial result we performed a paired significance analysis across independent generation seeds. For each seed, we measured the improvement in alignment from mean pooling relative to the best single token, and tested whether this improvement is consistently greater than zero across runs. Because the number of seeds is small, we use an exact one-sided sign-flip permutation test, which evaluates how likely it is to observe seed-level improvements consistent under the null hypothesis of no advantage for mean pooling. For both m-kNN and CKA on generated representations, all 5 seeds favor mean pooling, yielding an exact $p$-value of $1/32$. This provides initial evidence that the gain from mean pooling is a reproducible effect across runs.
>
> # 5. A fixed temperature / decoding strategy is used, and its impact is not explored.
>
> Thank you for raising this concern! We will incorporate your suggestion into our revision.
>
> # 6. The paper does not yet show clear downstream benefits, so the practical impact is still somewhat speculative.
>
> Thank you for the suggestion. We include this result in our revision by evaluating pooled representations on downstream tasks. We compare generated-token mean pooling against prompt-side mean pooling, prompt last-token pooling, and generation last-token pooling on tasks that assess whether representations preserve the semantic structure of the vision embedding space. Figure: https://imgur.com/a/v1eEc7q. Generated-token mean pooling performs best across all metrics, indicating that pooled representations are more useful for downstream tasks such as retrieval, ranking, and clustering.
> Overall, we clarify our impact. The immediate significance of our work is that it provides a new perspective for the representation learning and interpretability community to understand model behavior. In particular, recent work has shown strong interest in cross-modal convergence and alignment (e.g., “The Platonic Representation Hypothesis”), yet these works typically study representations as static embeddings. We instead study how representations evolve during generation, and show that viewing them as dynamic across tokens and layers can reveal structure in how models process information that is not visible from a static embedding perspective. We believe this perspective is especially relevant to understanding how and when such alignment emerges. For example, we observe phases such as recall, which raises the question of why the model appears to bring an internal world model into view during generation. This is an observation that would not be visible from single embeddings.
>
> —
>
> Thank you again for your helpful feedback! Do these answers address your concerns with the paper? If not, what further clarification or modifications could we make to improve your score?

---

> > ### Author Rebuttal · Reviewer_sEex · 2026-04-03
> >
> > Thank you for the detailed rebuttal and for providing additional experiments. The new results on longer-context behavior, shuffled-token controls, alternative reference models, and downstream evaluation substantially strengthen the paper and address many of my concerns.
> >
> > I still encourage the authors to phrase the strongest claims a bit more carefully, especially when moving from improved alignment to broader claims about grounding or semantic faithfulness. A clearer discussion of remaining limitations, such as dependence on reference models and decoding settings, would also improve the paper.
> >
> > Overall, the rebuttal has increased my confidence in the work, and I am keeping my current recommendation at weak accept.

---

> > > ### Author Response · Authors · 2026-04-05
> > >
> > > We agree with your suggestions, and will incorporate them into our revision. Thank you for the comments!

---

### Official Review · Reviewer_ZyKM · 2026-03-14

**Soundness:** 3
**Presentation:** 4
**Significance:** 3
**Originality:** 4
**Overall Recommendation:** 5
**Confidence:** 2

**Summary:**

The papers central question is to find a compressed representation of the sequence of hidden states generated by an AR model into a representation that accurately represents the internal states. The paper shows that pooling embeddings across all generated tokens consistently produces more informative and grounded representations than those derived from any individual token position. This is validated by measuring the alignment with external references across diverse domains, including vision-language, mathematical reasoning, and protein sequences.

**Compliance With Llm Reviewing Policy:**

Affirmed.

**Final Justification:**

The main review stands for the most part. I would also strongly suggest that the Impact Statement section of the paper be made strong by including all the points that the author's mentioned in their last response. As it stands, that section is weak. I am keeping my scores as they are high anyway.

**Key Questions For Authors:**

1. You find that prompt-side embeddings do not exhibit the same monotonic improvement or convex alignment structure as generated tokens. Can you comment on why the AR generative model treats the exact same sequence of tokens differently depending on whether they were part of the initial prompt or the model's own output?
2. Mean pooling is a low pass filter operation. This means it might remove the influence of transient tokens like "never", "not" etc., which would  completely change the "truth" of the generated sequence. Can you comment on this?
3. Your experiments show that the truth is distributed across time (all tokens), whereas there is no convex basin phenomenon across the layers. Can you elaborate on why that might be?
4. Your experiments are all conducted on modalities with some physical or structural properties (like images, protein structures). What do you think would happen to this distribution of the semantics on other domains like creative writing and subjective opinion? These applications don't have any objective reference models to rely on.
5. Could you elaborate on what the findings in your papers would mean for the wider research community? What impact would your findings?

**Limitations:**

yes

**Strengths And Weaknesses:**

+ The research problem of trying to understand the semantic content in AR generative models as a function of time (tokens) is interesting.
+ The rigorous, quantitative proof of how that information evolves and integrates during generation is novel.
+ Using representations from an independent reference models as a grounding factor is interesting.
+ Testing the observation on two metrics (CKA and m-KNN) is strong and increases the confidence that the phenomenon is probably not an artifact of the reference models.
- The summary could have contained more details that talks about what the main contributions of the paper are
- The Impact Statement could be made stronger. As it stands, the authors just hint at potentially significant impact without enumerating them.

---

> ### Author Rebuttal · Authors · 2026-03-30
>
> Thank you for taking the time to write a thoughtful review! We are encouraged that you find our research problem interesting and empirical analysis strong. We kindly respond to your concerns and number them for your convenience:
>
> # 1. Can you comment on why the AR generative model treats the exact same sequence of tokens differently depending on whether they were part of the initial prompt or the model's own output?
>
> We may have misunderstood your question, but in our setting, the prompt tokens and generated tokens are not the same sequence. The prompt-side representations are computed over the fixed input prefix, while the generated-token representations correspond to new tokens produced by the model conditioned on that prefix.
>
> We therefore compare two different parts of the autoregressive process: encoding the prompt versus generating a continuation. Generated tokens reflect the model’s evolving internal state, while prompt tokens come from a fixed input. Consistent with this, alignment improvements and convex structure appear during generation but not on the prompt side.
>
> # 2. Mean pooling is a low pass filter operation. This means it might remove the influence of transient tokens like "never", "not" etc., which would completely change the "truth" of the generated sequence.
>
> We agree that mean pooling does not preserve the semantics of each individual token. Instead, it produces a representation that is more aligned with the reference kernel, suggesting that averaging over many tokens yields a more coherent semantic summary of the underlying concept. This is interesting because it suggests that the relevant semantic information is not confined to individual tokens, but is shared across the trajectory. One possible interpretation is that these tokens span a common subspace, and averaging preserves the components that are consistent across them.
>
> # 3. Your experiments show that the truth is distributed across time (all tokens), whereas there is no convex basin phenomenon across the layers. Can you elaborate on why that might be?
>
> This is an interesting question! While our explanation is somewhat speculative, we believe it points to a useful direction for further study, as alignment across layers may provide insight into how representations evolve during processing.
>
> Our interpretation is that semantic information is distributed across tokens but not across layers in the same way. Across tokens, different positions capture complementary aspects of the underlying concept, so combining them improves alignment. Across layers, however, the model appears to progressively refine the representation for the decoding of an individual token, with later layers concentrating more task-relevant information.
>
> # 4. What do you think would happen to this distribution of the semantics on other domains like creative writing and subjective opinion?
>
> Our current results rely on domains with external structure, where alignment can be measured against an independent reference. As you have mentioned, in more subjective domains, such references are not well-defined so the same evaluation does not directly apply. However, we expect the temporal distribution of semantics to persist, as it arises from the generation process itself. Studying this would require alternative evaluators, such as representations derived from human preference, which we leave for future work.
>
> # 5. Could you elaborate on what the findings in your papers would mean for the wider research community?
>
> In our paper, we quantitatively study a scientific observation about the generated tokens of a language model. The immediate significance of our work is that it provides a new perspective for the representation learning and interpretability community to understand model behavior. In particular, recent work has shown strong interest in cross-modal convergence and alignment (e.g., “The Platonic Representation Hypothesis”), yet these works typically study representations as static embeddings. We instead study how representations evolve during generation, and show that viewing them as dynamic across tokens and layers can reveal structure in how models process information that is not visible from a static embedding perspective. We believe this perspective is especially relevant to understanding how and when such alignment emerges. For example, we observe phases such as recall, which raises the question of why the model appears to bring an internal world model into view during generation. This would not be observable from a single embedding.
>
> Overall, these findings suggest that generation-time representations are a useful object of study in their own right, and that analyzing trajectories of hidden states can provide a general framework for understanding model behavior.
>
> ---
>
> Thank you again for your helpful feedback! Do these answers address your concerns with the paper? If not, what further clarification or modifications could we make to improve your score?

---

> > ### Author Rebuttal · Reviewer_ZyKM · 2026-04-03
> >
> > I thank the authors for their clarifications. I would also strongly suggest that the answer to the question 5 be used to make the Impact Statement section of the paper strong. As it stands, that section is weak. I am keeping my scores as it is high anyway.

---

> > > ### Author Response · Authors · 2026-04-05
> > >
> > > We will incorporate your suggestions! Thank you for your response.

---

### Decision · Program_Chairs · 2026-04-30

**Decision:**

Accept (regular)

**Comment:**

This paper studies how to extract a single representation from the hidden-state trajectory of an autoregressive model during generation, and shows that mean pooling over generated tokens consistently outperforms any single token state in alignment with external reference representations.

The reviewers found this to be a solid and well-executed empirical contribution. The paper is clear, the central finding appears robust across multiple domains, and the rebuttal strengthened the submission further with longer-context results, additional reference models, comparisons to alternative pooling methods, statistical testing, and downstream evaluations.

Overall, I see this as a clear accept: useful and convincing empirical findings that should be of interest to the representation learning and interpretability communities.